

# Can the Mann model describe the typhoon turbulence?

Sara Müller[1,2], Xiaoli Guo Larsén[1], and Fei Hu[3]

[1]Department of Wind and Energy Systems, Danish Technical University, Risø Lab/Campus Frederiksborgvej 399, Roskilde 4000
[2]Sino-Danish Center for Education and Research (SDC), 100093, Beijing, China
[3]Institute of Atmospheric Physics, Chinese Academy of Sciences, 1000029 Beijing, China

**Correspondence:** Sara Müller (samul@dtu.dk)

**Abstract.** More and more wind farms are planned and built in regions prone to tropical cyclones. However, the current International Electrotechnical Commission (IEC) standard provides no clear guidelines on how to account for turbulence occurring during tropical cyclones. This study investigates how well the Mann turbulence model, a model referenced by the IEC, can model turbulence during tropical cyclone conditions. We analyzed sonic anemometer measurements at 60 m from four typhoon cases in the South China Sea. The Mann model was fit to the one-point spectra in different locations in the typhoon structure. We found that the Mann model can fit the observed spectra reasonably outside the typhoon eye and the rainbands. However, deficiencies are found regarding the following aspects: 1.) In the outer cyclone region, excessive energy is found at low wavenumbers smaller than $10^{-2.5}\,\mathrm{m}^{-1}$ associated with quasi-two-dimensional turbulence, 2.) In the inner cyclone and eyewall regions of one of the typhoons, we observe excessive energy at wavenumbers larger than $10^{-1}\,\mathrm{m}^{-1}$, 3.) Enhanced energy is found in the energy-containing subrange of the crosswind component in the inner cyclone region, and 4.) The energy level of the $uw$-cospectra is overestimated in the fitted Mann model. This study can serve as a baseline for further research addressing turbulence in tropical cyclones in the context of structural engineering.

## 1 Introduction

In recent years more and more wind farm projects are being planned and deployed in regions affected by tropical cyclones. These storms are associated with extreme wind speeds, which could damage civil structures. For tropical cyclone conditions, one of the turbine design parameters, $V_{ref}$, the 10-min mean value of the 50-year wind speed at hub height, has obtained some attention both in the International Electrotechnical Commission's standard (IEC) for wind turbines (IEC, 2019) and from researches (e.g., Ott, 2006; Larsén and Ott, 2022; Imberger et al., 2024). Recently, there have also been a few studies highlighting the importance of wind shear and veer during tropical cyclones in wind turbine load assessment, e.g., Sanchez Gomez et al. (2023) and Müller et al. (2024). At the same time, there have been a few studies investigating turbulence characteristics during Tropical Cyclone conditions, however, the studies of its impact on wind turbine loads are sparse.

Turbulence in tropical cyclones is influenced by different features in the wind field inherent to the tropical cyclone structure. The wind field varies over the various regions within the tropical cyclone, such as the eye, eyewall, and rainbands. For instance, in the eyewall, the vertical wind component is significant. Figure 1 illustrates the key regions inside a tropical cyclone. As a





tropical cyclone moves over wind turbines, the turbines are located in different regions over time. Within and between those regions, the wind characteristics often change. Therefore, the assumption of stationarity and horizontal homogeneity is not valid at all times (e.g., Huang et al., 2020).

There are only a few investigations on atmospheric turbulence models for tropical cyclone wind speed conditions. From turbulence measurements during the Coupled Boundary Layer Air–Sea Transfer (CBLAST) Hurricane experiment, Zhang

et al. (2011) calculated turbulence spectra in the outer cyclone region, away from the rainbands. The spectra had a similar form as the Kaimal model (Kaimal et al., 1972), however, energy was shifted to higher frequencies. This agrees with the findings from Li et al. (2012) based on measurements and large-eddy simulation (LES) modeling work by Worsnop et al. (2017). Li et al. (2012) further found excessive energy in the inertial dissipation range; this enhanced energy in these small eddies was argued to be related to the sea spray. Also, mesoscale flow structures can affect tropical cyclone turbulence. Li et al.

(2015) discussed the role of boundary layer rolls, horizontal elongated counter-rotating vortex structures, in tropical cyclone turbulence. Boundary layer rolls have been observed in various tropical cyclones (Zhang et al., 2008; Huang et al., 2018; Tang et al., 2021). Foster (2005) and later Gao and Ginis (2014) showed how boundary layer rolls form in idealized tropical cyclone simulations. Such roll structures affect turbulent fluxes in the tropical cyclone boundary layer (Zhu et al., 2010; Tang et al., 2021), which can cause the turbulence behaviors to deviate from classical turbulence modeling. Worsnop et al. (2017) noted

that enhanced coherence apparent in their LES could be related to these roll structures. In areas most interesting for wind energy, close to the coast, the turbulence wind field can be further complicated by the land and internal boundary layers (He et al., 2022).

For both turbine design and operation, it is crucial to know how a tropical cyclone affects the loads on turbines. To simulate wind turbine loads, the turbulence wind field needs to be modeled in a four-dimensional space around the turbine (Dimitrov

et al., 2017). Worsnop et al. (2017) analyzed turbulence based on LES and concluded that adjustments in wind turbine standards may be needed to capture the characteristics of turbulence in tropical cyclones. The IEC standard recommends using the Kaimal (Kaimal et al., 1972) or the Mann uniform shear model (Mann, 1994) (hereafter referred to as the Mann model). The Mann model is based on rapid distortion theory (RDT), assuming that turbulence eddies are elongated under linear wind shear over their lifetime. The Mann model is a well-established method within wind energy, yet it has not been thoroughly tested

for tropical cyclone conditions; Whether the RDT is a good basis for modeling turbulence in such conditions needs to be questioned. Notably, it has been documented by De Maré and Mann (2014) that the Mann model cannot adequately describe the large-scale fluctuations in the power spectra of the wind components and fails to account for mesoscale fluctuations.

Research on the mesoscale fluctuations for non-tropical cyclone conditions dates back to the 1970s, see Larsén et al. (2016) for an overview. It was proposed by Kim and Adrian (1999), and later discussed by Högström et al. (2002), that the interaction

between mesoscale fluctuations and the typical, three-dimensional boundary layer turbulence might be inactive. This theory was confirmed, demonstrated, and developed into a full-scale turbulence model, first by Larsén et al. (2016) using measurements from two stations, from the surface to 100 m high, and later in Larsén et al. (2021) from another Met station with measurements up to 245 m high. There have been several follow-up studies on this topic worldwide, and one of them, Cheynet et al. (2018), used the FINO 1 data for more detailed spectral analysis for different stability conditions. Following Cheynet et al. (2018),





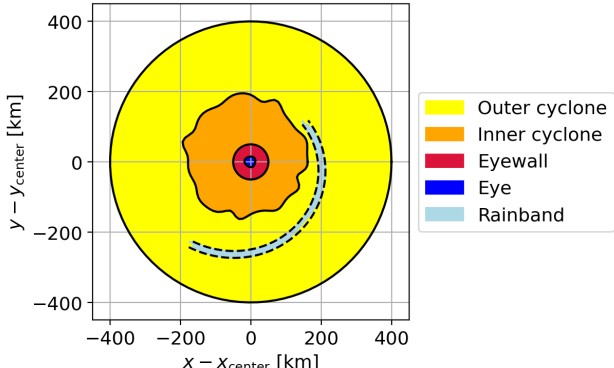

**Figure 1.** Illustration of typhoon regions, as defined for the analysis.

Syed and Mann (2024) also used the FINO 1 data and proposed an approach to extend the Mann model to lower wavenumbers by manually adding large-scale fluctuations. Mesoscale turbulence was shown to be significant in the presence of organized atmospheric features, such as open cells and fronts (Larsén et al., 2019). This suggests that mesoscale turbulence may also be important during tropical cyclones.

These findings suggest that the Mann model may face substantial limitations when applied to tropical cyclone conditions. Although Larsen et al. (2016) fitted the Mann model to turbulence spectra derived from data between the outer rainbands of two hurricanes (Zhang et al., 2011), it has never been systematically addressed how well the Mann model can fit tropical cyclone turbulence. In particular, it is not clear how well the Mann model can describe the turbulence behavior in different tropical cyclone regions, and how large the contribution of mesoscale fluctuations is during tropical cyclone conditions. In this study, we investigate tropical cyclone turbulence in different cyclone regions by analyzing high-frequency measurements of four typhoons in the South China Sea. In particular, we assess where the Mann model can fit tropical cyclone turbulence and it has limitations.

In the following, the measurements used for the analysis are introduced in Section 2, where the methods for data processing and for calculating the power spectra are introduced. The Mann model and the fitting are also explained in this section. Results are presented in Section 3, followed by Discussions and Conclusions in Sections 4 and 5.

# 2 Data and Methods

## 2.1 Data

We use measurement data from four typhoons to study turbulence spectral behaviors. These are typhoon Hagupit in 2008, typhoon Nuri in 2008, typhoon Prapiroon in 2006, and typhoon Chanchu in 2006. Table 1 summarizes the mast locations, the height of the analyzed data, instruments, and sampling frequencies of all measurements. The locations of the masts are also shown in Figure 2, with Figure 2b-e as close-up view at the four locations and their surroundings. Under the assumption that



**Table 1.** Summary of the measurement data during four tropical cyclones.

| Name | Observation site | Coordinates | Measurement height | Measurement device | Frequency |
|------|------------------|-------------|--------------------|--------------------|-----------|
| Hagupit | Zhizai island | 111.38°E / 21.45° N | 60 m | Gill, WindMaster Pro | 10 Hz |
| Nuri | Delta island | 113.71°E / 22.14° N | 60 m | Gill, WindMaster Pro | 10 Hz |
| Prapiroon | Xuwen | 110.12°E / 20.24°N | 60 m | Campell, CSAT3 | 4 Hz |
| Chanchu | Red Bay | 115.60°E / 22.74°N | 60 m | Campell, CSAT3. | 4 Hz |

the wind field at 60 m is predominately influenced by the surface over a 6 km area, land-dominated fetches were identified. The corresponding wind directions are indicated by red shading in Figure 2b-e.To make the best use of the limited measurement data, we analyze measurement periods with wind direction from both land-dominated and water-dominated sectors. The fetch effect is addressed in the analysis.

The measurements of typhoon Hagupit are described in detail in Li et al. (2012); they were also analyzed in two other studies, Liu et al. (2011) and Wenchao et al. (2011). The three-dimensional wind speed was measured using a Gill WindMaster Pro ultrasonic anemometer at 10 Hz. The device is mounted at 60 m on the Bohe meteorological tower located on the Zhizai island in the Guangdong Province (Fig. 2b). The island is approximately 120 m in length and 50 m in width, with a distance of about 4.5 km to the southwest of mainland China. The tower base is 11 m above sea level. To the west and southwest, the

wind is fairly exposed to open waters. To the northwest, mainland China affects the wind at the measurement height.

     The measurements used to analyze typhoon Nuri are described in Li et al. (2015). The three-dimensional wind field was measured by a Gill WindMaster Pro ultrasonic anemometer at 60 m at a 10 Hz frequency. The measurement mast stands on the Delta Island, at the entrance of the Pearl River estuary (Fig. 2c). The tower base is 93 m above sea level. The wind is fairly exposed to open waters. Smaller islands are within 6 km distance from the Delta island. Sectors from affected directions are

labeled as mixed fetch in the following analysis. The city Macau is more than 10 km to the northeast of the measurement tower.

     The wind field used for the analysis of typhoon Prapiroon was measured at 4 Hz by a CSAT3 ultrasonic anemometer from Campell. The ultrasonic anemometer is installed at 60 m on a meteorological tower in Xuwen, in the southwest of the Guangdong Province (Fig. 2d). The tower is located in the coastal region facing the Qiongzhou Strait to the south. The distance to the Sea is around 200 m to the west and 700 m to the south and southeast. In these directions, the tower is exposed to winds

from the sea. In the northern sectors, the wind comes from the land mass.

     Typhoon Chanchu's wind field was measured by a CSAT3 ultrasonic anemometer from Campell at 4 Hz. The anemometer is installed at 60 m on a meteorological tower on the Red Bay peninsula in Shanwei, northern Guangdong Province (Fig. 2e). The Red Bay peninsula faces the South China Sea to the southeast. The distance to the shore is around 500 m to the south and east and 1.5 km to the north. To the west, the wind comes from the land, while to the other wind directions, the wind comes

from the water.

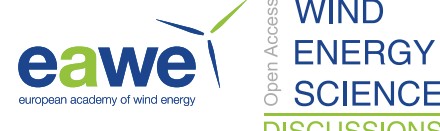

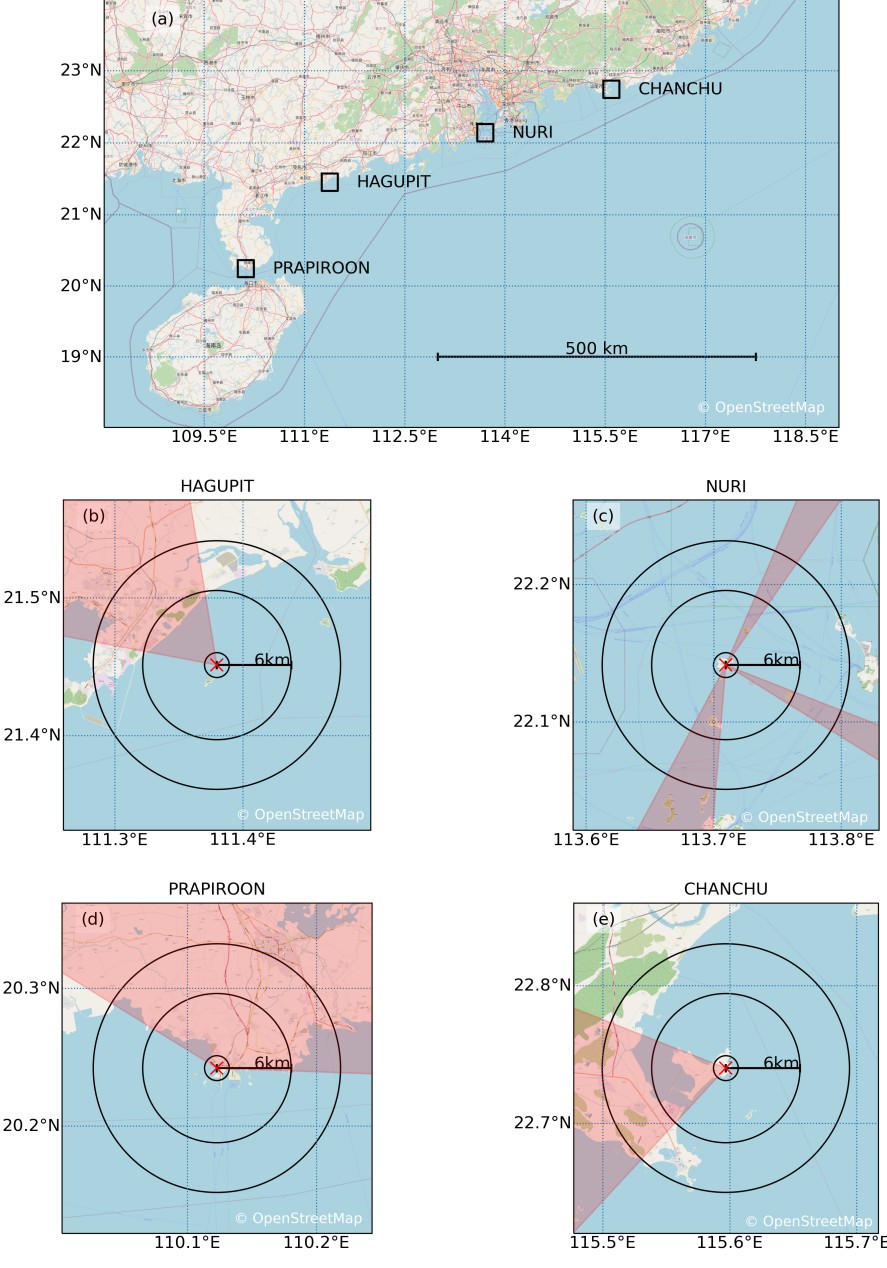

**Figure 2.** Measurement locations during four tropical cyclones. Panel a) gives a large-scale overview, and panels b-e) feature the detailed environment given by the black rectangles in panel a). The red cross in panel b-e) indicates the station location. Black circles give distances of 1, 6, and 10 km. The red shading indicates wind directions, where the innermost 6 km are influenced by land. (adapted from OpenStreetMap)



### 2.1.1 Data preprocessing

Tropical cyclones are associated with strong wind speeds, large waves, and intense rain. The severe conditions can cause technical difficulties for measurements and hence affect the data quality. One of the consequences is that there are frequent spikes in the wind measurements. To reduce uncertainties related to data quality, we process the data in four steps:

1. Filtering of spikes: Spikes are filtered as described in Vickers and Mahrt (1997), see their Sect. 6a. Measurement points are characterized as spikes if a measured wind component differs more than 3.5 standard deviations from a five-minute running average over more than three consecutive time steps. The parameters chosen in Vickers and Mahrt (1997) resulted in a good spike detection for the measurements analyzed in this study. To make the best use of the data, we discard only data with more than 10 % spikes.

2. Filtering of repeating values: For typhoon Chanchu, the measurements of the three wind components feature episodes with consecutive repeated values. We checked the data for episodes where one of the three components does not change for longer than 10 s and removed intervals with more than 5% of these repeated values in any wind component.

3. Filtering of flickering: For typhoons Hagupit, Chanchu, and Prapiroon, there are episodes where individual wind-component measurements flickered between large and small values. To detect these episodes, the difference between
consecutive measurement values is calculated. The data is removed when more than 5 % of these differences are larger than $3\,\mathrm{m\,s^{-1}}$ within one minute in any wind component.

4. Filling the data gaps: Continuous data series are needed to calculate time spectra. Therefore gaps in the data are filled. Gaps from removed spikes are linearly interpolated. Disregarded data due to repeated or fluctuating values are filled using data from a time window of 15 minutes before to 15 minute after the center of the data gap. Only segments with
less than 15% filled data are used to calculate spectra.

### 2.1.2 Power spectra calculation

To calculate the wind power spectra, the time series is analyzed in 30-minute segments. In each 30-minute segment, the wind field is decomposed along the mean horizontal wind vector into the alongwind component $u$, the crosswind component $v$, and the vertical wind component $w$. This is only possible if the wind direction does not change significantly during the segment.
Here, we only use 30-minute segments where the maximal change in the wind direction is less than $20°$, based on ten-minute moving averages. For each 30-minute segment, the power spectrum in the frequency domain is calculated using Fast Fourier transform. To reduce the influence of non-stationarity, the time series of $u$, $v$, and $w$ are linearly detrended before calculating the spectrum. The Taylor hypothesis is used to transform the spectra from the frequency domain to the wavenumber domain through:

$$k = 2\pi \cdot f/U, \tag{1}$$





where $k$ is the wavenumber, $f$ the frequency, and $U$ the average wind speed over the 30-minute segment. To reduce spectral noise and to better assess the spread of the Mann parameters subsequently obtained through the fitting (see Sect. 2.2), we use overlapping 30-minute segments to calculate the spectra. The center times of all consecutive 30-minute segments are shifted by 5 minutes, leading to a 25-minute overlap between consecutive segments. The individual spectra based on these 30-minute

segments are smoothed by averaging the values of the normalized spectral power in bins of $log_{10}(k)$, with 10 bins per decade in $k$. The median spectrum is then calculated by taking the median of the normalized spectral power across the different spectra for each $k$-bin.

## 2.2 The Mann model and the fitting

The Mann model applies RDT on a von Kármán isotropic tensor (von Kármán, 1948). Here the main assumptions include

a constant vertical shear and neutral atmospheric stability. RDT allows assessing how the wind shear acts on a wind field represented as a superposition of Fourier modes. Intuitively, the effect of the wind shear can be understood by considering its effect on individual turbulent eddies. The shear elongates the eddies in the mean shear direction. Mann (1994) extended RDT by assuming a characteristic non-dimensional lifetime ($\Gamma$) of turbulent eddies as a function of wavenumber. $\Gamma$ controls the total stretching of the eddies in a steady state. To illustrate how $\Gamma$ affects the spectra, Fig. 3 shows normalized $u$, $v$, $w$ spectra ($kF_u$,

$kF_v$, $kF_w$) and $uw$-cospectra ($kF_{uw}$) for different $\Gamma$-values. For a $\Gamma$ value of zero, the turbulence is isotropic. With increasing $\Gamma$, the energy in the $kF_u$ and, to a lesser extent, in $kF_v$ and $kF_{uw}$ increases, and the energy in $kF_w$ decreases. At the same time, the peak wavenumber $k_p$ changes. It becomes smaller for the $u$ and $v$ components and larger for the $w$ component. Note that the spectral energy increases more for the $u$ component than for the $v$ component. Also, the difference between the $k_p$ for the $u$ and $v$ components increases. A key advantage of the Mann Model is that spatial coherence and $uw$-cospectra are incorporated

into the model.

In addition to $\Gamma$, the Mann Model has two more adjustable parameters: a turbulent length scale ($L$), and a parameter controlling the decay of turbulence, $\alpha\epsilon^{2/3}$. $L$ is the size of the largest energy-containing eddies, and it controls $k_{max}$. In $\alpha\epsilon^{2/3}$, $\alpha$ is the Kolmogorov's constant, and $\epsilon$ the turbulent dissipation rate. $\alpha\epsilon^{2/3}$ modulates the magnitude of the spectra and is related to the standard deviation of the wind components (Mann, 1994).

We estimate the three Mann model parameters by finding the fit of the Mann model to the spectra obtained from measurements that have the minimal difference between the normalized autospectra of the three velocity components and the normalized $uw$-cospectra. Similar to Syed and Mann (2024), we use a downhill simplex algorithm (McKinnon, 1998) to minimize the error function ($F_{error}$):

$$F_{error} = \sum_{i=1}^{3}\sum_{j=1}^{N} |k_j \cdot F_{i,Mann}(k_j) - k_j \cdot F_{i,meas.}(k_j)| + \sum_{j=1}^{N} |k_j \cdot F_{13,Mann}(k_j) - k_j \cdot F_{13,meas.}(k_j)| \tag{2}$$

Here, $F_{i,Mann}$ and $F_{i,meas}$ (i=1,2,3) are the autospectra of the $u$, $v$ and $w$ components of the Mann model, and the measured spectra, respectively. $F_{13}$ is the $uw$-cospectrum. $N$ is the number of wavenumbers in the spectra after the logarithmic averaging. In order to get the best model fit in the energy-containing range of the spectra, only $k < 0.1$ m$^{-1}$ are used for the fit. In order to avoid extreme values of these parameters that are not physically sensible, when fitting the Mann model parameters,



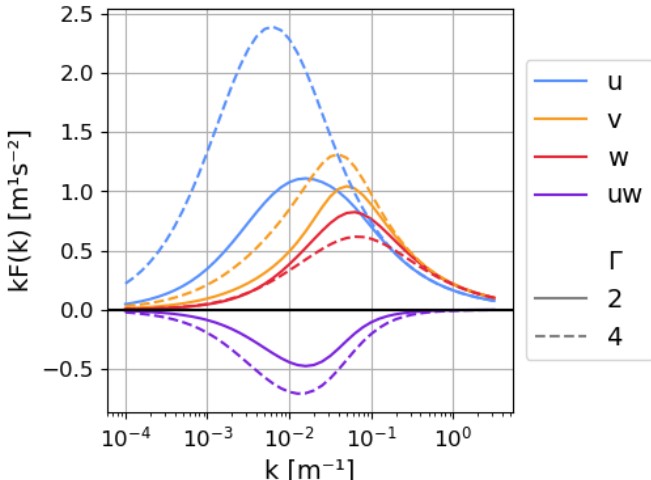

**Figure 3.** Spectra obtained by the Mann turbulence model with different $\Gamma$ values. Here L is 30 m and $\alpha\epsilon^{2/3}$ is 1 $\mathrm{m^{3/4}s^{-2}}$.

we limit the parameter range according to values usually found in atmospheric conditions in the literature: $L \in [0, 300]$ m, $\alpha\epsilon^{2/3} \in [0, 2]$ $\mathrm{m^{2/3}}$, and $\Gamma \in [0, 5]$.

### 2.3 Analysis of turbulence behaviors in various typhoon regions

The wind properties vary greatly between different regions in typhoons, be it the eye, eyewall, rainbands, or outer region. Therefore we define regions with similar wind field characteristics, on which we base the analysis of the turbulence spectra. We distinguish between five regions as illustrated in Fig. 1. These regions are: 1.) The outer cyclone region, 2.) the inner cyclone region, 3.) the eyewall, defined as a radially narrow band including the strongest horizontal wind speeds, 4.) the eye, a region with low to moderate horizontal wind speeds, radially inward of the eyewall, and 5.) the rainbands. To distinguish between these regions, we use the mean wind speed from the measurements (WS) (ten-minute running average at 60 m measurement height), the distance to the cyclone center (R), and the radius of the maximal wind speed (RMW). Both R and RMW are obtained from the International Best Track Archive for Climate Stewardship (IBTrACS) (Gahtan et al., 2024; Knapp et al., 2010) and linearly interpolated to the measurement time. The spectral behaviors are not analyzed for the eye and rainband regions because the wind direction changes significantly in these regions, which makes the Fourier transform invalid. The boundaries of the outer cyclone, inner cyclone, and eyewall regions are set roughly based on the following criteria:

- The outer cyclone region includes measurements where $R \lesssim 400$ km and $WS \lesssim 17$ $\mathrm{m\,s^{-1}}$. We take $R = 400$ km as the outer boundary of the outer cyclone region because the distance of the outermost closest isobar is roughly 400 km for the four analyzed typhoons, according to the IBTrACS.

- The inner cyclone region includes measurements with $WS \gtrsim 17$ $\mathrm{m\,s^{-1}}$ and $R \gtrsim 2$ RMW.





– The eyewall includes measurements where R $\lesssim$ 2 RMW. Note that eyewall asymmetries cannot be addressed by this criteria. The inner edge of the eyewalls is taken where WS $\simeq 25 \text{ m}^{-1}$, based on visual inspection of the measured wind speed time series.

Flexibility in these criteria is allowed, such that the time series can be divided into continuous periods belonging to the same region. We further distinguish between the front and the back sectors of the typhoon, i.e. the measurement period before and after the timestep where the cyclone center is closest to the measurement station, a similar approach to Li et al. (2012).

For typhoon Prapiroon, a rainband is identified by using an index of enhanced fluctuations in the vertical wind, as will be shown in Sect. 3.1. This approach was proven to be successful when verified by a satellite-based precipitation dataset, which shows consistent coverage of rainband with a region of large vertical wind fluctuations. Specifically, we use the Integrated Multi-satellite Retrievals for Global Precipitation Measurement Mission (IMERG) (Huffman et al., 2023). This is a global, gridded precipitation dataset with a resolution of 30 minutes and $0.1°$, available from 1988 to the present. The IMERG data was obtained by the integrated Multi-satellite retrieval algorithm v7, which inter-calibrates, merges, and interpolates measurements, including satellite microwave precipitation estimates, microwave-calibrated infrared satellites, and precipitation gauge analyses. The dataset has previously been used successfully to characterize tropical cyclone precipitation (Rios Gaona et al., 2018). Note that other regions than the rainband are also affected by precipitation and convection, though to a lesser extent.

## 3 Results

### 3.1 Meteorological background of the typhoons

We first give a short overview of the four typhoons to help understand how these typhoons influenced the wind fields and turbulence spectra during different periods and, thereof, different typhoon regions. The measured horizontal wind speed, direction, and vertical wind speed are shown in Figs. 4-7 (subplots on the left panel) alongside the typhoon track (subplot on the right panel) for the four typhoons, respectively. Periods analyzed in more detail are marked in blue and labeled for reference, with the label "FO" for Front Outer cyclone, "FI" for Front Inner cyclone, "FEW" for Front Eyewall, "BI" for Back Inner cyclone, "BO" for Back Outer cyclone and "BEW" for Back Eyewall, both in the left and right panels.

Typhoon Hagupit's track over the measurement mast is shown in Fig. 4g; also shown is the precipitation data from IMERG at 2.30 UTC on 4. August. Hagupit formed over the Northwest Pacific in September 2008; see Fig. 4g. Hagupit moved westwards and entered the South China Sea through the Luzon Strait. Around 00 UTC on 24 September, Hagupit made landfall on mainland China at Dianbai, close to the measurement station. The passing of the typhoon shows clearly in the measured wind field (Fig. 4a-c). At 06 UTC on 23. September, the station was around 450 km to the northwest of the cyclone center. The peak in the wind speed at around 9 UTC on 23. September, is most likely associated with a rainband, showing also in the IMERG dataset (not shown). After the passing of the rainband, the wind speed increased, while the typhoon approached the measurement location. In the inner cyclone region, both horizontal and vertical wind speeds increased gradually over time.





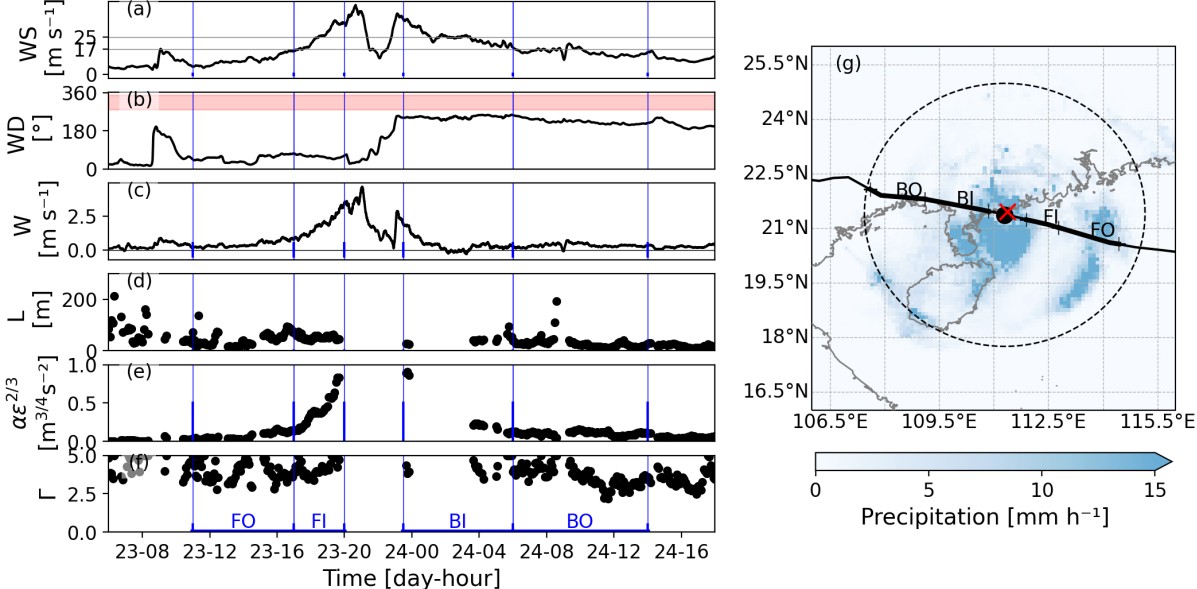

**Figure 4.** a-c) Measured wind field at 60 m during typhoon Hagupit (ten-minute running average): a) horizontal wind speed, b) wind direction (red shading indicates wind directions, where the innermost 6 km are influenced by land), c) vertical wind speed, d-f) fitted Mann parameters, g) typhoon track (black solid line, from west to east), measurement station (red cross), precipitation at 22 UTC on 23. September from IMERG (shading), cyclone center at that time (black point), and 400 km distance to cyclone center (black dashed line). Typhoon regions further analyzed are marked in panels f) and g): front outer cyclone (FO), front inner cyclone (FI), back inner cyclone (BI), and back outer cyclone (BO).

Between about 20 and 24 UTC, the front eyewall, the eye, and the back eyewall passed the measurement location. The passage

of the eye is visible in the drop in the horizontal and vertical wind speed (Fig. 4a and c). The passing of the eye is further marked by a drastic change in wind direction from northerly in the front to southerly in the backside of the cyclone center. The two eyewalls, front and back, are marked by peaks in the horizontal and vertical wind speed. Because of the significant changes in the wind direction and the presence of many spikes (hence high uncertainty in data quality) in the measurements, it becomes invalid to use the Fourier Transformation, and therefore, data in the eyewalls and the eye regions of typhoon Hagupit

are not further analyzed. After the passing of the back eyewall, the cyclone center moved further away from the station, and the measured horizontal and vertical wind speeds decreased with time.

The measurements during typhoon Nuri show a similar pattern with increasing wind speeds towards the front and back eyewall, separated by a drop in the wind speeds (see Fig. 5). In contrast to Hagupit, Nuri's center/track did not move directly over the measurement location (see Fig. 5g). Accordingly, the drop in the wind speed is less drastic than during typhoon Hagupit.

At the closest point to the center, the station was at the inner flank of the eyewall (at 10 UTC on 22. August 2008). The wind speed around this time is decreased compared to the eyewall winds before and after it. However, as the cyclone center stayed northward of the station, the wind direction did not reverse. At the same time, the measurement time series covers a longer



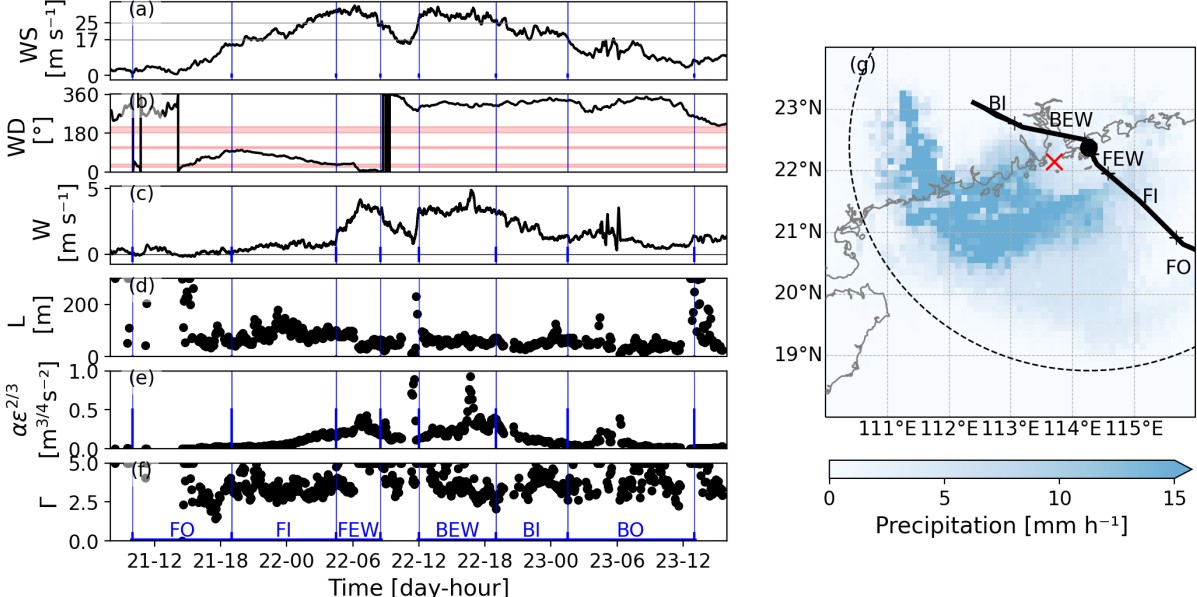

**Figure 5.** a-c) Measured wind field at 60 m during typhoon Nuri (ten-minute running average): a) horizontal wind speed, b) wind direction (red shading indicates wind directions, where the innermost 6 km are influenced by land), c) vertical wind speed, d-f) fitted Mann parameters, g) typhoon track (black solid line, from southwest to northeast), measurement station (red cross), precipitation at 23 UTC on 16. August from IMERG (shading), cyclone center at that time (black point), and 400 km distance to cyclone center (black dashed line). Typhoon regions further analyzed are marked in panels f) and g): front outer cyclone (FO), front inner cyclone (FI), front eyewall (FEW), back eyewall (BEW), back inner cyclone (BI), and back outer cyclone (BO).

period in the eyewalls and inner cyclone region than for Hagupit. The horizontal and vertical winds fluctuated around 4 UTC on 23. August. This may be associated with the presence of convective cells inside the typhoon. However, precipitation is not
as clearly indicated as in the case of Hagupit in the IMERG dataset (not shown).

As the track is far away from the site (about 160 km), measurements during typhoon Prapiroon don't contain information about the eye, shown as a lack of the drop of wind speeds between two eyewalls (Fig. 6). Between 12 UTC on 2. August and 18 UTC on 3. August 2006 the horizontal and vertical wind speed at the measurement location gradually increased with time,
while the typhoon center moved closer to the station. From 18.30 UTC on 2. August onward, the wind field is characterized by fluctuations in the wind speed, wind direction, and between up and downdrafts. We assume, that these fluctuations are produced by convection and raincells within a rainband. This is confirmed by the rain in the IMERG dataset over the station at 2.30 UTC (Fig. 6 d).





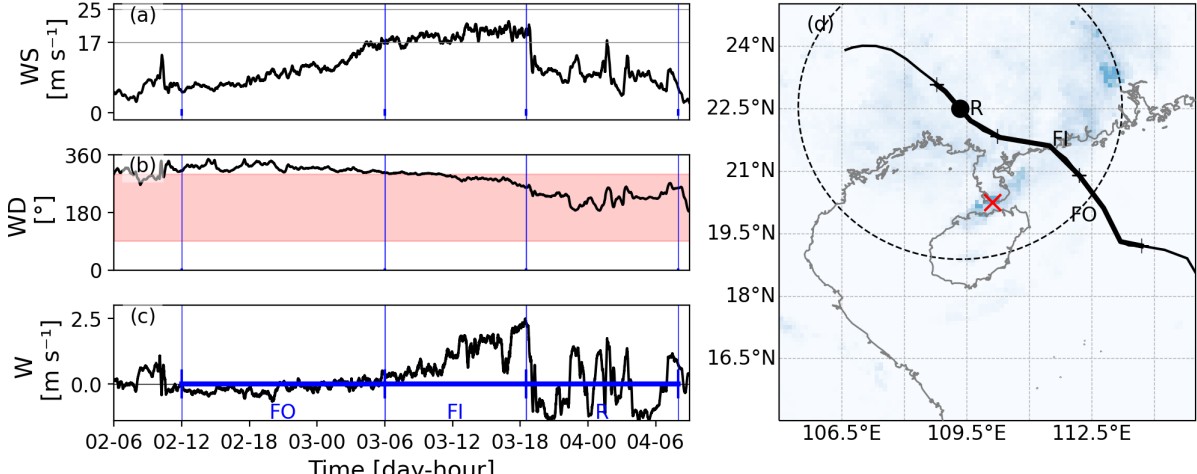

**Figure 6.** a-c) Measured wind field at 60 m during typhoon Prapiroon (ten-minute running average): a) horizontal wind speed, b) wind direction (the red shading indicates wind directions, where the innermost 6 km are influenced by land), c) vertical wind speed, d) Typhoon track (black solid line, from southwest to northeast), measurement station (red cross), precipitation at 2.30 UTC on 4. August from IMERG (shading), cyclone center at that time (black point), and 400 km distance to cyclone center (black dashed line). Typhoon regions further analyzed are marked in panels c) and d): front outer cyclone (FO), front inner cyclone (FI), and rainband (R).

Typhoon Chanchu approached the station from the southwest in May 2006 (see Fig. 7). Similar to the case of Prapiroon, the long distance between the measurement station and the typhoon track results in the absence of eye information in the measurements. The measured horizontal wind speed is largest around 10 UTC on 17. May. This was when the cyclone center was closest to the measurement station, which is around 100 km southeast of the station. A large part of the measurements contain prolonged periods with constant values for wind speed, which indicates poor data quality and the data are therefore not suitable for analyzing turbulence. Yet, two periods with good measurement quality could still be identified and extracted. The first period is in the inner cyclone region at 9 UTC on 17. May, when the cyclone center was about 120 km to the southeast of the measurement mast. This time step is shown in Fig. 7d. The second period, at around 21 UTC, covers the outer cyclone region after the landfall of typhoon Chanchu.

## 3.2 The Mann Parameters

For those 30-minute periods of measurements that fulfill the criteria described in Sect. 2.1.1, we calculate the spectra and fit the Mann model parameters, as described in Sect. 2.1.2 and Sect. 2.2. Here, we first describe the parameter range we obtained by the fitting and how it varies between the tropical cyclone cases and regions. Thereafter, we discuss the goodness of the fit and special features of the spectra. The resulting Mann model parameters for typhoon Hagupit and Nuri are shown in Fig. 4 and Fig. 5 in subplots d, e, and f, for $L$, $\alpha\epsilon$, and $\Gamma$, respectively. The obtained parameters are further summarized in Table 2 for all





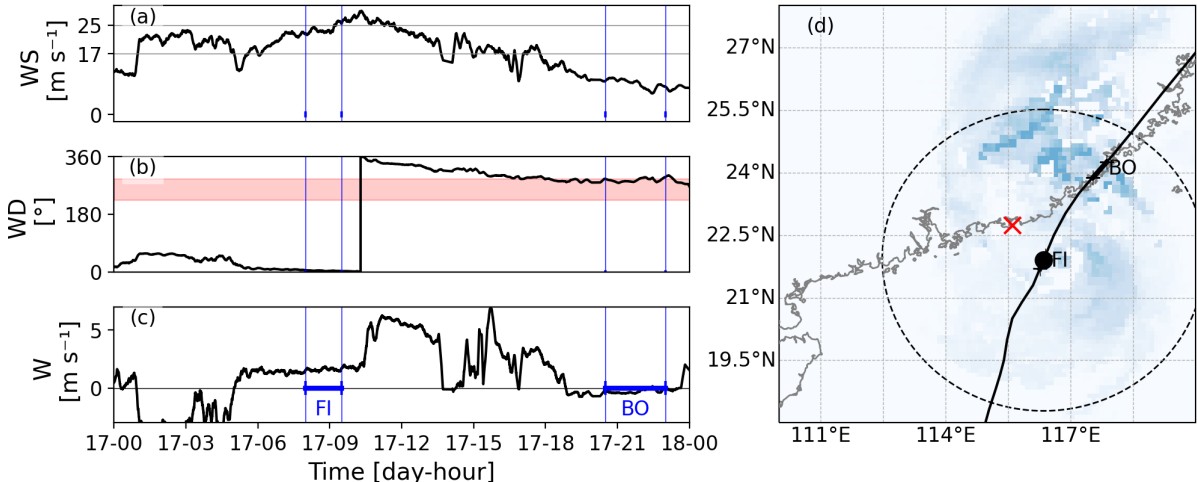

**Figure 7.** a-c) Measured wind field at 60 m during typhoon Chanchu (ten-minute running average): a) horizontal wind speed, b) wind direction (red shading indicates wind directions, were the innermost 6 km are influenced by land), c) vertical wind speed, d) typhoon track (black solid line, from southwest to northeast), measurement station (red cross), precipitation at 9 UTC on 17. May from IMERG (shading), cyclone center at that time (black point), and 400 km distance to cyclone center (black dashed line). Typhoon regions further analyzed are marked in panels c) and d): front inner cyclone (FI) and back outer cyclone (BO).

four typhoons. Therein we list the median values of these parameters for the specified tropical cyclone regions (see Sect. 2.3) and the sample size per region.

First, we examine the range of the $L$ parameter, which correlates with the size of the largest eddies. Across the four typhoons, the median $L$-values obtained in the different regions range from 26.3 to 85.9 m (see Table 2). $L$ is largest for typhoon Nuri. Note that the measurements for typhoon Nuri were taken from a tower with a base height of 93 m, placing the ultrasonic

anemometer at 153 m above sea level. This absolute height is larger than that of the other three typhoon cases, which could explain the larger $L$-values from Nuri compared to the other typhoons. In all four typhoon cases, the $L$-values in the front sectors before the storm center passed the station are larger than those in the corresponding back sectors. However, the overall $L$-values from these four typhoons are comparable to those obtained in non-typhoon conditions during neutral stratification at similar measurement heights in Høvshore and Østerid (Denmark) (Peña et al., 2010a; Peña, 2019).

For the four typhoon cases, $\alpha\epsilon^{2/3}$ varies between different cyclone regions (see Fig. 4e and 5e). The smallest $\alpha\epsilon^{2/3}$-values are obtained from the spectra in the outer cyclone region, and the largest $\alpha\epsilon^{2/3}$-values are obtained from those of the inner cyclone region and eyewalls, suggesting more efficient dissipation at stronger winds. This variation in $\alpha\epsilon^{2/3}$ is related to the horizontal wind speed, where $\alpha\epsilon^{2/3}$ increases with increasing wind speed, a relation consistent with Sathe et al. (2013). In their study they find $\alpha\epsilon^{2/3}$ on the order of 0.03 $\mathrm{m^{3/4}\,s^{-2}}$ for wind speeds of 7 $\mathrm{m\,s^{-1}}$, and 0.13 $\mathrm{m^{3/4}\,s^{-2}}$ for wind speeds of 16 $\mathrm{m\,s^{-1}}$

at 90 m during neutral atmospheric stability in Høvshore (Denmark). The median values obtained during the four typhoons range between 0.02 $\mathrm{m^{3/4}\,s^{-2}}$ in Nuri's front outer cyclone region, and 0.50 $\mathrm{m^{3/4}\,s^{-2}}$ in Chanchu's front inner cyclone region.





**Table 2.** Summary of the different analyzed regions for the four typhoon cases. "Fetch" indicates whether the wind comes from the Land or from the Sea, and "Numb. samples" gives the number of samples analyzed in each region. Median values are given for the wind speed (WS), and the Mann model parameters $L$, $\alpha\epsilon^{2/3}$, and $\Gamma$.

| Region | Typhoon | Sector | Fetch | Numb. samples | $WS$ [m s$^{-1}$] | $L$ [m] | $\alpha\epsilon^{2/3}$ [m$^{4/3}$ s$^{-2}$] | $\Gamma$ |
|---|---|---|---|---|---|---|---|---|
| Outer | Hagupit | Front | Sea | 59 | 8.9 | 39.4 | 0.05 | 3.8 |
| Outer | Hagupit | Back | Sea | 88 | 14.6 | 26.3 | 0.11 | 3.6 |
| Outer | Nuri | Front | Sea | 47 | 7.0 | 68.8 | 0.02 | 3.1 |
| Outer | Nuri | Back | Sea | 98 | 10.9 | 41.7 | 0.04 | 3.8 |
| Outer | Prapiroon | Front | Sea | 183 | 9.9 | 26.7 | 0.19 | 2.8 |
| Outer | Chanchu | Back | Land | 27 | 9.0 | 24.9 | 0.06 | 3.3 |
| Inner | Hagupit | Front | Sea | 33 | 24.8 | 53.1 | 0.37 | 4.0 |
| Inner | Hagupit | Back | Sea | 19 | 21.6 | 41.6 | 0.21 | 4.3 |
| Inner | Nuri | Front | Sea | 114 | 20.3 | 85.9 | 0.06 | 3.3 |
| Inner | Nuri | Back | Sea | 60 | 20.3 | 54.1 | 0.11 | 3.5 |
| Inner | Prapiroon | Front | Land | 77 | 18.7 | 52.9 | 0.18 | 3.3 |
| Inner | Chanchu | Front | Sea | 15 | 23.4 | 38.3 | 0.50 | 2.7 |
| Eyewall | Nuri | Front | Mixed | 44 | 29.5 | 66.5 | 0.22 | 5.0 |
| Eyewall | Nuri | Back | Sea | 81 | 28.1 | 57.2 | 0.24 | 3.5 |

The $\alpha\epsilon^{2/3}$-values are smaller for typhoon Nuri, compared to the other typhoon cases at corresponding wind speeds. This is likely because, the measurements of Nuri are at an effective height of 153 m, compared to about 60 m for other typhoon cases (see previous paragraph), and for surface-driven turbulence $\alpha\epsilon^{2/3}$ and the dissipation rate usually decrease with height, see

Fang et al. (2023), and also e.g., Peña (2019) for non-typhoon conditions.

The median $\Gamma$-values range between 2.7 and 5 across the analyzed regions and typhoons, with a median of 3.5 (see Table 2). This range is comparable to values found in non-typhoon conditions. For instance, the range for $\Gamma$ is between 3 and 3.5 for neutral conditions at heights between 60 and 100 m at the Høvshore and Østerid site (Sathe et al., 2013; Peña, 2019). Note that we set an upper boundary of 5 to the $\Gamma$ range allowed when fitting the Mann model (see Sect. 2.2). In the fitting, larger values

could, in principle, be obtained. However, for such large values their physical meaning is unclear, and it is recommended not to apply the Mann model beyond this value (Private communication with Jakob Mann, the creator of the Mann model).




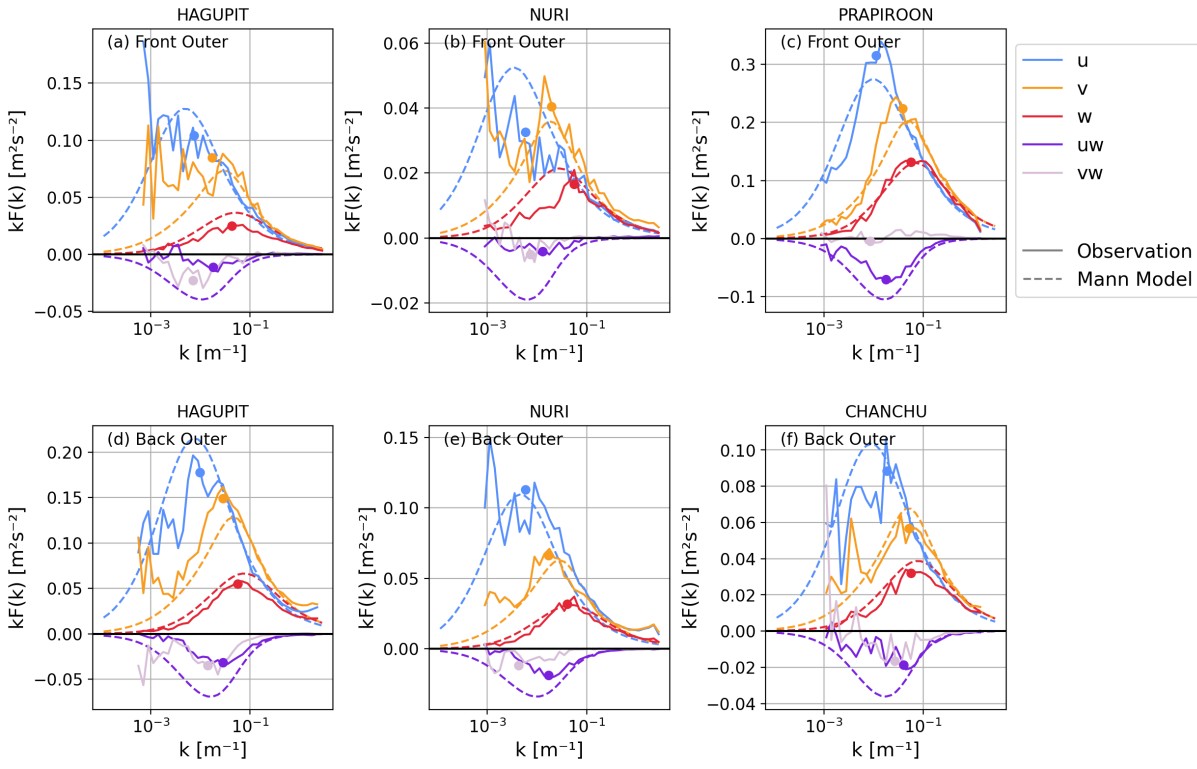

**Figure 8.** Spectra from the front and back outer cyclone regions of the four typhoon case: Median of observed spectra (Solid lines), fitted Mann Model (dashed lines), and estimated maxima of the observed spectra (points).

### 3.3 How well does the Mann model fit the spectra

In this section, we summarize when the Mann Model can describe the measured spectra and what features it cannot cover. For that, we compare spectra obtained from measurements and the Mann model fit. Figure 8, 9, and 10 show the normalized

spectra as a function of wavenumber for the outer cyclone, the inner cyclone, and the eyewall region, respectively. Note that there are a number of 30-minute segments for each case (listed in Table 2), and hence the same amount of spectra and sets of the Mann model parameters. Here, we show the median of both the measured spectra (solid curves) and the Mann model (dashed curves). The corresponding median Mann model parameters can also be found in Table 2.

The spectra from measurements resemble the Mann model in several aspects, listed below. This is true not only for the outer

region (Fig. 8), but also the inner region (Fig. 9), and eyewall region (Fig. 10). This means that some features covered by the Mann model are, in most instances, also evident in the measured spectra. These features include 1.) that the normalized spectra have a maximum at a wavenumber $k_p$ and the normalized spectral energy generally decreases towards larger and smaller wavenumbers, 2.) $kF_u$ is larger than $kF_v$ and $kF_w$ at $k < k_p$, 3.) $kF_{uw}$ is negative, which means that the momentum flux of the $u$ wind component is downward, and 4.) $k_p$ is smaller for the $u$ component than that for the $v$ component and largest for



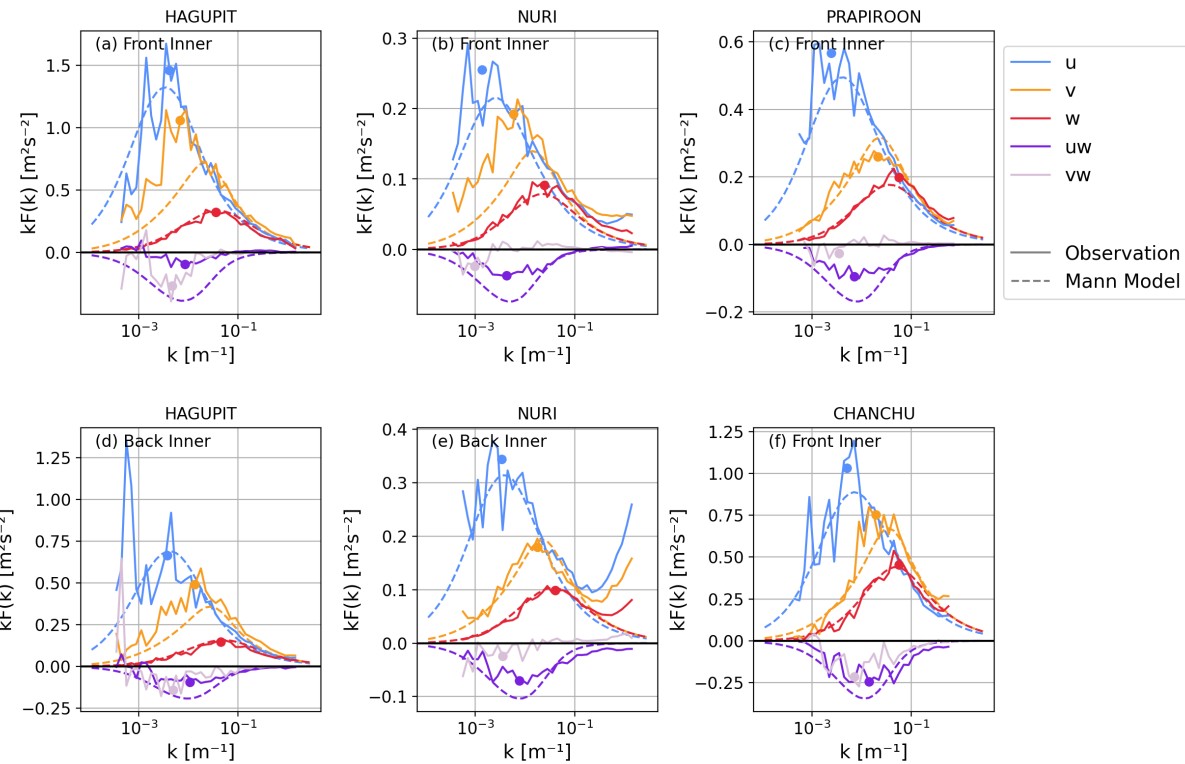

**Figure 9.** Spectra from the front and back inner cyclone regions of the four typhoon case: Median of observed spectra (Solid lines), fitted Mann Model (dashed lines), and estimated maxima of the observed spectra (points).

the $w$ component. These features are consistent with the usual boundary-layer spectra for wind components. However, when zooming in on these features, there are discrepancies between the model and measurements.

First, we address the slope of the spectra at $k > 10^{-1}$ m$^{-1}$. This wavenumber range is often referred to as the inertial subrange. In this range, the slope of the normalized spectra on a double logarithmic axis is expected to be $-2/3$ in classical boundary layer theory and in the Mann model. Table 3 lists the slope at $k > 10^{-1}$ m$^{-1}$ as $slope_{k1}$. The slope is obtained from a linear fit to the median of the measured spectra for the different regions and typhoons. For typhoon Hagupit, Prapiroon, and Chanchu, the slope at $k > 10^{-1}$ m$^{-1}$ is mostly close to or slightly larger than $-2/3$. Differently, in typhoon Nuri's inner cyclone region and eyewall region, the slope is significantly larger at $k > 10^{-1}$ m$^{-1}$. This can be seen clearly in Fig. 9e and Fig. 10a,b, where the observed spectra show excessive energy at the $k \gtrsim 10^{-1}$ m$^{-1}$. Using the same measurements, Li et al. (2015) also reported the positive slope in the spectra during typhoon Nuri. They proposed that the secondary peak in the spectra at these large frequencies (or large wavenumbers, respectively) may be related to the evaporation of sea spray and the associated buoyancy force, as well as coastal roughness changes and surface waves. Also, other studies reported on enhanced spectral energy at high frequencies during typhoon conditions (Tao and Wang, 2019; He et al., 2022). The enhanced energy at $k \gtrsim 10^{-1}$ m$^{-1}$ could also be caused by flow distortions from the meteorological tower (Barthlott and Fiedler, 2003). This





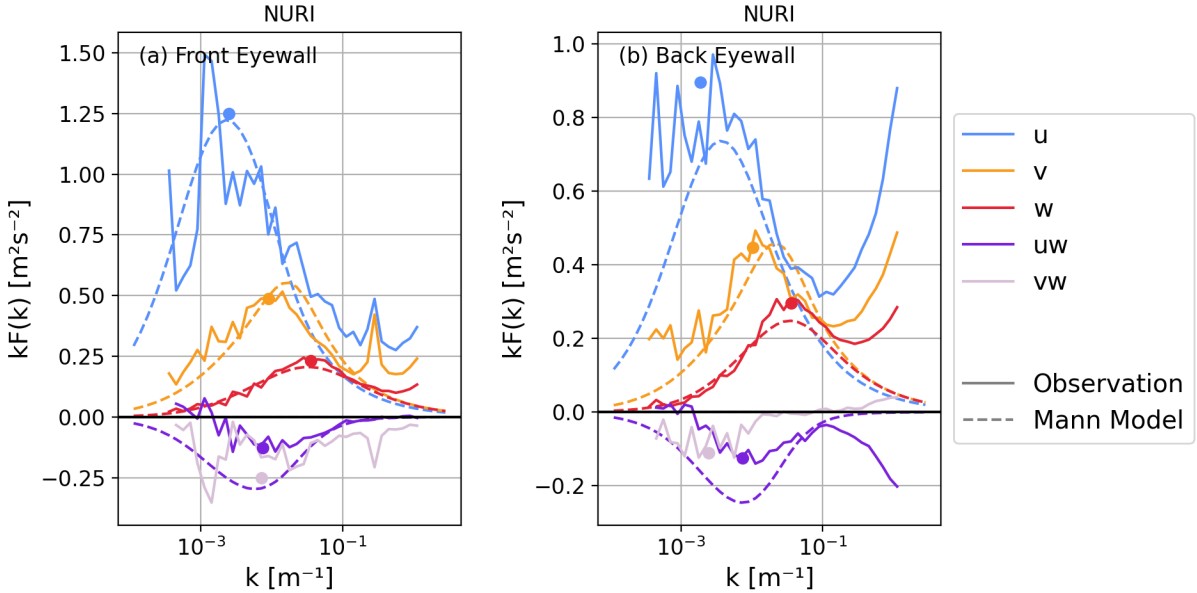

**Figure 10.** Spectra from the front and back eyewall of typhoon Nuri: Median of observed spectra (Solid lines), fitted Mann Model (dashed lines), and estimated maxima of the observed spectra (points).

spectral behavior is, however, only obvious during Nuri, particularly in the back inner and back eyewall regions, and therefore, it can not be generalized. More measurements and research are needed to understand the spectral behaviors at $k > 10^{-1}$ m$^{-1}$ where excessive energy is present.

Second, we investigate the importance of large-scale fluctuations in tropical cyclone turbulence. To do this, we examine the slope of the spectra for $k < 10^{-2.5}$ m$^{-1}$, which is on the large-scale side of the approximate peak wavenumber. The slopes from the observed $u$ and $v$ spectra at $k < 10^{-2.5}$ m$^{-1}$ are listed in Table 3 as $slope_{k2}$. The corresponding slope of the $u$ and $v$ components from the Mann model fitted to the spectra is on the order of 0.6 (not shown). In five out of six cases in the outer cyclone regions, the slope in the observed spectra is 0.01 or less. This is evident in Fig. 8a, b, d, e, and f, where there is enhanced energy at small wavenumbers, e.g., $k \simeq 10^{-3}$ m$^{-1}$ in the observed spectra compared to the Mann model. The small slope in these spectra suggests the presence of significant large-scale wind fluctuations that are beyond the concept of the Mann model. We discuss this feature in more detail in Sect. 4. The slope at $k < 10^{-2.5}$ m$^{-1}$ is mostly larger in the inner cyclone and eyewall regions shown in Figs. 9 and 10 compared to the outer cyclone region.

Third, we examine one of the key spectral parameters, the peak wavenumber $k_p$ and the corresponding energy $k_p F(k_p)$ in Figs. 8-10, from both the measurements and the Mann model. For the Mann model curve, it is straightforward to find $k_p$ and the corresponding $k_p F(k_p)$. For measurements, we decide $k_p$ through visual inspection, as there is often enhanced and fluctuating $kF(k)$ for $k < 10^{-3}$ m$^{-1}$, which makes it difficult to find an objective $k_p$. For reference, the obtained maxima are marked as points in Figs. 8-10 together with the median values of the spectra. Table 3 lists the obtained $k_p F(k_p)$ for the





measurements and Mann model, respectively. In most cases, $k_p F_u(k_p)$, $k_p F_v(k_p)$, and $k_p F_w(k_p)$ from the fitted Mann model are close to that of the measured spectra. However, particularly in the inner cyclone region (Figs. 9), there are cases where the Mann model significantly underestimates $k_p F_v(k_p)$. This is clearly visible in Fig. 9a,b, and d, which show the front inner cyclone region of Hagupit and Nuri and the back inner cyclone region of Hagupit. In these three cases, the Mann model seems
to have captured the $v$-variability in smaller eddies, while it missed out on the low wavenumber variability. At the same time, $k_p$ of the $v$ component is overestimated in the Mann model, or in other words shifted to larger wavenumbers, compared to the measurements. This is not the case for the $k_p$ of the $u$ component, for which, the Mann model did an overall much better job. In other words, the dominating eddy sizes for $u$ and $v$ are significantly closer to each other in measurements than the Mann model suggests. We summarize this quantity as $log(k_{p,v}/k_{p,u})$ in Table 3. From the values, it is evident that this feature is consistent
in 13 out of the 14 observed spectra in the different regions and typhoons. This agrees with the findings of Zhang et al. (2011): They found that $k_p$ of the $u$ and $v$ components are closer to each other in the spectra observed in tropical cyclones than in the turbulence models from Kaimal et al. (1972) and Miyake et al. (1970). The multiple peaks in the spectra suggest the presence of multi-scale active eddies in the atmosphere, making it challenging for a simple model fit; e.g., in Fig. 8, the Mann model captures one of the dominant modes of turbulence, and missed out the rest, most of which are from large scales.
Fourth, we find that the Mann model overestimates the magnitude of $kFuw$ for $k \leq k_p$. This is evident in Figs. 8-10, as well as from the estimated $k_p F_{uw}(k_p)$, listed in Table 3. Such an overestimation of the $kF_{uw}$ has also been observed over Denmark, i.e., in areas not affected by tropical cyclones (Peña et al., 2010b; Peña, 2019). Non-negligible negative values are found in $kF_{vw}$ in 10 out of the 14 regions, which is another feature that differs significantly from the classical turbulence behaviors.

## 4  Disussion

Turbulence during tropical cyclone conditions is a less studied topic, especially in the context of wind energy(Kosovic and et al., 2025, e.g.,). However, it is becoming increasingly important with the rapid development of wind energy in cyclone-prone regions. Accurate estimation of turbulence is important for safer operation and design of wind turbines. This study investigates how well the Mann model describes typhoon turbulence and what limitations the model may have. The Mann model is one of the turbulence models recommended by the IEC standard that can be used to simulate four-dimensional wind fields, as is
necessary, for example, for load calculations. The study is conducted by analyzing sonic anemometer measurements from four stations where four typhoons from 2006 and 2008 passed by, with the eye of one of the typhoons passing over the station, and the eyes of the other typhoons passing within a few hundred kilometers of the respective stations.

While the collection of these data is far from rich, it offers several opportunities. First, although the Mann model is well established in the wind energy sector, it has not been tested with actual measurements for tropical cyclone conditions. To our
knowledge, this is the first study of the Mann model using tropical cyclone measurements. Second, the data analyzed include situations where the measurement stations experience a variety of typhoon structures, here called "regions" and illustrated in Fig. 1, including the eye, eyewall, rainband, inner cyclone, and outer cyclone regions. Thus, this study addresses the turbulence spectral behavior in different parts of a typhoon and the corresponding Mann model performance.





**Table 3.** Comparison between median measured spectra (Mes.) and the fitted Mann model (Mann). The $slope_{k1}$ gives the mean slope of $kF_u$, $kF_v$ and $kF_w$ at $k > 10^{-1}\mathrm{m}^{-1}$, and the $slope_{k2}$ gives the mean slope of $kF_u$, and $kF_v$ at $k < 10^{-2.5}\mathrm{m}^{-1}$. The columns $k_pF_u(k_p)$, $k_pF_u(k_p)$, $k_pF_u(k_p)$, $k_pF_{uw}(k_p)$ give the normalized spectral power at the peak wavenumber $k_p$ for $u$, $v$, $w$, and $uw$ wind component respectivly. The $log(\frac{k_{p,v}}{k_{p,u}})$ compares the $k_p$ of the $u$ and $v$ component.

| Region | Typhoon | Sector | $slope_{k1}$ | $slope_{k2}$ | $k_pF_u(k_p)$ $[\mathrm{m}^2\,\mathrm{s}^{-2}]$ | | $k_pF_v(k_p)$ $[\mathrm{m}^2\,\mathrm{s}^{-2}]$ | | $k_pF_w(k_p)$ $[\mathrm{m}^2\,\mathrm{s}^{-2}]$ | | $k_pF_{uw}(k_p)$ $[\mathrm{m}^2\,\mathrm{s}^{-2}]$ | | $log(\frac{k_{p,v}}{k_{p,u}})$ | |
|---|---|---|---|---|---|---|---|---|---|---|---|---|---|---|
| | | | Mes. | Mes. | Mes. | Mann | Mes. | Mann | Mes. | Mann | Mes. | Mann | Mes. | Mann |
| Outer | Hagupit | Front | -0.6 | -0.14 | 0.10 | 0.13 | 0.08 | 0.07 | 0.02 | 0.04 | -0.01 | -0.04 | 0.36 | 0.84 |
| Outer | Hagupit | Back | -0.4 | -0.18 | 0.18 | 0.21 | 0.15 | 0.13 | 0.06 | 0.07 | -0.03 | -0.07 | 0.46 | 0.84 |
| Outer | Nuri | Front | -0.6 | -0.57 | 0.03 | 0.05 | 0.04 | 0.04 | 0.02 | 0.02 | -0.00 | -0.02 | 0.52 | 0.73 |
| Outer | Nuri | Back | -0.3 | -0.18 | 0.11 | 0.11 | 0.07 | 0.06 | 0.03 | 0.03 | -0.02 | -0.03 | 0.46 | 0.84 |
| Outer | Prapiroon | Front | -0.8 | 0.27 | 0.32 | 0.27 | 0.22 | 0.20 | 0.13 | 0.13 | -0.07 | -0.10 | 0.54 | 0.73 |
| Outer | Chanchu | Back | -0.5 | 0.01 | 0.09 | 0.10 | 0.06 | 0.07 | 0.03 | 0.04 | -0.02 | -0.04 | 0.44 | 0.73 |
| Inner | Hagupit | Front | -0.7 | 0.56 | 1.46 | 1.32 | 1.06 | 0.72 | 0.32 | 0.33 | -0.10 | -0.39 | 0.22 | 0.84 |
| Inner | Hagupit | Back | -0.5 | 0.21 | 0.66 | 0.69 | 0.49 | 0.36 | 0.15 | 0.16 | -0.10 | -0.19 | 0.56 | 0.84 |
| Inner | Nuri | Front | -0.2 | 0.33 | 0.26 | 0.22 | 0.19 | 0.14 | 0.09 | 0.08 | -0.04 | -0.07 | 0.64 | 0.74 |
| Inner | Nuri | Back | 0.2 | 0.01 | 0.34 | 0.31 | 0.18 | 0.19 | 0.10 | 0.10 | -0.07 | -0.10 | 0.72 | 0.84 |
| Inner | Prapiroon | Front | -0.5 | 0.59 | 0.57 | 0.49 | 0.26 | 0.32 | 0.20 | 0.18 | -0.09 | -0.17 | 0.94 | 0.74 |
| Inner | Chanchu | Front | -0.4 | 0.69 | 1.03 | 0.89 | 0.75 | 0.67 | 0.46 | 0.44 | -0.24 | -0.35 | 0.58 | 0.74 |
| Eyewall | Nuri | Front | -0.1 | 0.55 | 1.25 | 1.22 | 0.49 | 0.55 | 0.23 | 0.21 | -0.13 | -0.30 | 0.56 | 0.84 |
| Eyewall | Nuri | Back | 0.3 | 0.05 | 0.90 | 0.74 | 0.45 | 0.46 | 0.30 | 0.25 | -0.12 | -0.25 | 0.74 | 0.84 |

We note that the data cover only four typhoons. Due to the limited number of typhoons covered in this study, and further complicated by the data quality and the division into regions based on the cyclone structure, the number of samples for each region is severely limited. Therefore, the results of this study cannot be generalized for turbulence characterization for the different regions within a typhoon. Nevertheless, the data were valuable in showing some distinct typhoon-related features and in addressing the applicability of the Mann model to such conditions.

Our analysis provided important insights into the variability at the lower wavenumbers ($k \lesssim 10^{-2.5}\ m^{-1}$) within typhoons. In particular, we found in Sect. 3.3 that both $u$ and $v$ can have excessive energy at lower wavenumbers during tropical cyclones, especially in the outer cyclone region, see Fig 8. This is clearly shown by the smaller slope of the spectra at $k < 10^{-2.5}\ m^{-1}$ compared to the Mann model (Table 3). The large energy content at these lower wavenumbers is related to the significant mesoscale fluctuations observed in previous studies dealing with turbulence under non-tropical cyclone conditions (De Maré





and Mann, 2014; Cheynet et al., 2018; Larsén et al., 2019, eg.). One of the implications of the significant energy contribution from mesoscale fluctuations is that using a 30-minute window of the time series, may not be long enough; this is consistently suggested in all six subplots of Fig. 8, as there is a high energy level at the lowest wavenumber, even for the relatively well-behaving spectra from the front outer region of typhoon Prapiroon. In the spectra of the outer cyclone region, the width of the energy-containing range is broad, thus containing a broad scale range of active turbulence contributors. This width is

considerably narrower in the inner cyclone and eyewall regions shown in Figs. 9 and 10, likely due to the better-defined convection-driven turbulence in these regions. Nonetheless, the high energy level at the lowest wavenumber also suggests that the 30-minute time series is not long enough for turbulence estimation.

  To further understand the sources of variability at the lower wavenumbers, we examined the role of boundary layer rolls, which are often present in tropical cyclones (Foster, 2005; Zhang et al., 2008; Tang et al., 2021). The turbulence behavior

associated with the boundary layer rolls was studied by Smedman (1991) for a case over the Baltic Sea using turbulence measurements from a 145 m meteorological tower. They found excessive spectral energy in the $v$ and $w$ components at low frequencies (around 0.003 Hz) and countergradient $v'w'$-fluxes. Excessive spectral energy in the $v$-component was also observed in three of the four typhoons in our analysis. To see if this is related to the boundary layer rolls or other organized structures, we simulate the four typhoons with the Weather Research and Forecasting (WRF) model (version 4.51) (Skamarock

et al., 2019a). The simulation uses four one-way nested domains, with a horizontal grid spacing of 18, 6, 2, and 2/3 km, respectively. Please refer to the appendix A for more details on the simulation setup. It should be noted that a spatial resolution of 2/3 km in the WRF simulation does not resolve all the wind variability associated with structures such as the boundary layer rolls. Nevertheless, the model output is expected to be helpful in identifying the presence or absence of the associated patterns. We observe the presence of the roll-like patterns in the simulations of typhoon Hagupit and Nuri close to the corresponding

measurements. Roll patterns are also present during Chanchu and Prapiroon, but they are far away from the measurement station during the analyzed time periods.

  To illustrate these patterns, Fig. 11a and b show the simulated wind field from the innermost model domain at a height of 60 m, at 3 UTC on 22. August 2008 during Typhoon Nuri. Panel b provides a closer look at the area around the measurement site (marked in red). In this area, the boundary layer rolls can be seen as the elongated bands of alternating above- and

below-average horizontal wind speeds, circulating the cyclone center. Near the measurement site, the rolls are in a northwest-southeast orientation, see Fig. 11b. At this time, the cyclone flow from the north has passed the upwind land areas, which modified the wind field. The vertical wind speed has a similar structure to the horizontal wind, with elongated bands of upward and downward motion, respectively (not shown). To demonstrate the excessive energy associated with these organized wind features, we compute and compare the wind speed power spectra from regions with and without roll structure (see Appendix

A for details). The spectra for the area with rolls and for an area without visible boundary layer rolls are shown in Fig. 11c. The spectrum from the area with rolls shows enhanced energy over all scales, especially in the range from $k \sim 5 \cdot 10^{-5}$ to $8 \cdot 10^{-4}\ \mathrm{m}^{-1}$, corresponding approximately to a wavelength of 1 to 20 km. Our model setup with the finest spatial resolution of 2/3 km can only resolve wind variability up to about 5 km effective resolution (Skamarock, 2004). Therefore the simulations do not capture the strength of the variability for $k > 2 \cdot 10^{-4}\ \mathrm{m}^{-1}$. In addition, Svensson et al. (2017) found that the simulated





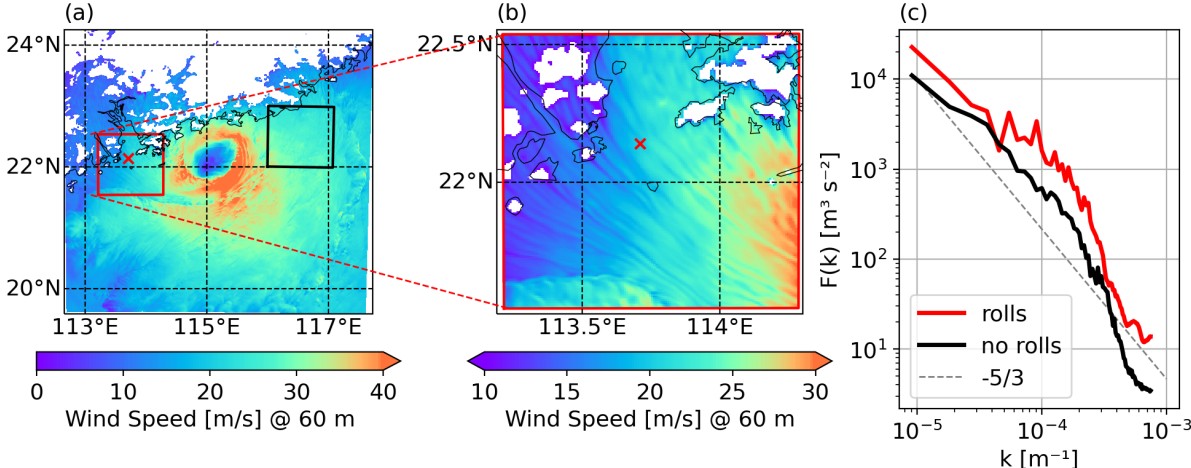

**Figure 11.** a,b) Simulated horizontal wind speed during typhoon Nuri at 60 m above sea level. Panel b shows a detailed view of the area marked in red in panel a; note the different color scales. The red cross in a and b gives the measurement location, and the black contours give the coastlines. White areas have a terrain elevation larger than 60 m. c) Wind speed power spectra calculated from the simulated wind speed at 60 m from the areas marked in red and black in panel a, representing areas with and without boundary layer rolls. The dashed line gives a slope of -5/3 for reference.

roll wavelength in the mesoscale simulation decreases with increasing grid resolution. In tropical cyclones, boundary layer roll structures are found to have wavelengths on the order of 0.6-1.5 km based on Synthetic Aperture Radar (SAR) (Morrison et al., 2005; Zhang et al., 2008; Huang et al., 2018).

To conclude the discussion on the role of organized atmospheric features, the simulations show that boundary layer rolls were present in Nuri's front inner region (Fig. 11), in Hagupit's front outer cyclone region, and in Hagupit's front inner cyclone region (not shown). The corresponding spectra all show enhanced energy in the crosswind component at $k \gtrsim 10^{-1.5}$ ms$^{-1}$ compared to the Mann model (Fig. 8a, and Fig. 9a,b). This supports the hypothesis that the excessive energy in $kF_v(k)$ at low wavenumber could be associated with the roll-like structures inside the typhoons. However, more research with more data is needed to further support this hypothesis.

As a final part of the discussion, we need to address the uncertainties associated with the spectral analysis of the measurements, apart from the small data samples mentioned earlier in this section. During the passage of a typhoon, the changes in both wind speed and direction can be significant, challenging the validity of the stationarity assumption. This is also one of the reasons why we limited the length of the time series to 30 minutes, and no longer. These changes in wind speed and direction challenge the analysis in terms of: 1.) decomposing of the wind vector into the along wind and crosswind components, 2.) using of the Taylor Frozen hypothesis to convert spectra from the frequency domain to the wavenumber domain, and 3.) taking the average spectra of the regions defined in Sect. 2.3. Limitations on these three points are discussed in the following.





We have assessed whether the decomposition of the wind vector into its $u$ and $v$ components is successful or not, by calculating and comparing the spectra of $u$ with that of the wind speed. A successful decomposition would result in more or less the same spectra from the two. This has been our case, which gives us credit to the analysis of the $u$ and $v$ components
of the turbulence spectra. Note that we did not analyze the data in the eye for typhoon Hagupit due to the non-stationary time series.

Current aeroelastic codes for wind turbine load assessment require wind field input in the spatial domain. A wavenumber spectrum can serve this purpose better than a frequency spectrum. Since spatial, continuous measurements are almost non-existent, we use point measurements. Consequently, we need to convert the spectra from the frequency to the wavenumber
domain using the Taylor Frozen hypothesis, implicitly assuming a constant advection velocity. Traditionally, aeroelastic simulations are run for 10-minute periods. The use of 10 minutes for atmospheric surface layer turbulence, along with the constant mean velocity, is usually a reasonable assumption. However, such an assumption is highly questionable when a tropical cyclone is passing by. In this case, we need to use longer time series to include larger eddies, especially since modern wind turbines are larger than the atmospheric surface layer. This could lead to uncertainties in the transformed wavenumber spectra regarding
the energy density at the corresponding wavenumbers.

As described in Sect. 2.3, our analyses are mainly based on the median value of the spectra obtained within defined typhoon regions. Due to the large-scale typhoon structure, the wind and turbulence properties are expected to be associated with these regions. Therefore, the analysis is expected to be sensitive to the definition of each region. Given the complex spatial structure of a constantly evolving and moving tropical cyclone, our definition may be an oversimplification. Furthermore, when comparing
the median spectra from measurements with the Mann model, we used the median of the $L$, $\alpha\epsilon^{2/3}$, and $\Gamma$ parameters. In general, these parameters are different from those obtained by fitting the Mann model to the median spectra.

In addition, spatial coherence also has a known impact on wind turbine loads (Dimitrov et al., 2017). The coherence resulting from the Mann model depends on the $L$ and $\Gamma$ parameters. However, it is beyond the scope of the current study to address the tropical cyclone coherence and to assess how well the Mann model can capture it. Further studies are needed for such an
investigation.

## 5    Conclusions

We analyzed one-point power spectra during four typhoons in the South China Sea, namely the typhoons Chanchu, Prapiroon, Hagupit, and Nuri. The data were sorted into different regions inside a typhoon structure: outer region, inner region, eyewall, eye, and rainbands. We fitted these spectra to the Mann uniform shear model and assessed how well the model describes the
spectral behaviors during tropical cyclone conditions in different cyclone regions.

The Mann model can fit the tropical cyclone power spectra to some degree, mostly in the outer cyclone region, and the model is most challenged in the eye and rainbands. Discrepancies between the observed spectra and the model were found mainly regarding the following points:





1. In the outer cyclone region, the slope of the observed spectra at $k < 10^{-2.5}$ m$^{-1}$ is, for three out of four typhoon cases, significantly smaller than in the Mann model. This corresponds to a rather broad energy-containing range with multiple peaks and a flatter top, suggesting the presence of large-scale wind fluctuations. The Mann model can capture one of the peak modes and misses most of the large-scale fluctuations.

2. In the inner cyclone region and eyewalls of typhoon Nuri, excessive spectral energy is found at $k > 10^{-1}$ m$^{-1}$. This is argued in Li et al. (2015) to be related to sea spray evaporation, surface roughness changes, and waves. This is not captured in any classical turbulence models, including the Mann model.

3. In the inner cyclone region, significantly enhanced spectral energy is found in the crosswind component of typhoon Nuri and Hagupit at $k \lesssim 10^{-2}$ m$^{-1}$ compared to the Mann model. At the same time, the peak frequencies of the $u$ and $v$ spectra are closer to each other than the Mann model suggests.

4. For all typhoons and regions, the energy in the $uw$-cospectra is overestimated in the Mann model fitted to the observed spectra at $k \lesssim 10^{-2}$ m$^{-1}$.

5. Non-negligible energy was found in the observed $vw$-cospectra, which cannot be modeled by the Mann uniform shear model.

As listed above, one of the striking characteristics of typhoons is the spectral behaviors related to the crosswind component $v$. The current study addresses this by relating it to the often-present boundary rolls inside tropical cyclones using both measurements and numerical simulation. Our study suggests a positive linkage between the excessive energy in the $v$ component and the presence of boundary-layer rolls.

We acknowledge that turbulence is highly variable during tropical cyclones and is case-dependent. Further studies based on larger datasets should be conducted for more robust results. This study can serve as a reference when investigating tropical cyclone conditions, particularly for structural engineering purposes.

*Code and data availability.* The IBTrACS dataset is available at https://doi.org/10.25921/82ty-9e16 (Gahtan et al., 2024), the IMERG dataset at https://doi.org/10.5067/GPM/IMERG/3B-HH/07 (Huffman et al., 2023), the WRF source code at https://doi.org/doi:10.5065/D6MK6B4K (Skamarock et al., 2019b), name lists for the WRF simulations at https://doi.org/10.5281/zenodo.14610013 (Müller, 2025), and the ERA5 dataset at https://doi.org/10.24381/cds.bd0915c6 (Hersbach et al., 2018).

## Appendix A: Simulation setup and calculation of spectra from simulation

Simulations of the four typhoons are performed using WRF version 4.51, to analyze the flow structure at and around the measurement locations of the four typhoons. WRF is run on four one-way nested domains, d01, d02, d03, and d04. The outermost domain is d01. The nested domains d02, d03, and d04 are run in a vortex following grid configuration. The horizontal grid



spacing is 18, 6, 2, and 2/3 km for d01, d02, d03, and d04 respectively. For the four simulations, d04 has $769 \times 769$ horizontal grid points. 70 vertical model levels are used, with the lowermost levels at heights around 8, 26, 47, 72, 102, 139, 183, and

292 m. Initial and boundary conditions are taken from the ERA5 reanalysis data (Hersbach et al., 2018), and the sea surface temperature is taken from OSTIA (Donlon et al., 2012). The simulations are started at 12 UTC on 21. August for typhoon Nuri 2008, at 0 UTC on 23. September 2008 for typhoon Hagupit, at 12 UTC on 16. May 2006 for typhoon Chanchu, and at 0 UTC on 2. August 2006 for typhoon Prapiroon. The first 6 simulation hours after the simulation start are used for the model spinn-up. The following parametrization schemes are used: The Yonsei University boundary layer scheme (YSU) (Hong et al.,

2006), the revised MM5 surface layer scheme (Jiménez et al., 2012) with the isftcflx option 2, the Thompson microphysics scheme (Thompson et al., 2008), the RRTMG radiation scheme (Iacono et al., 2008), and the Kain-Fritch cumulus scheme (Kain, 2004). The latter is only activated in d01.

To analyze the simulated wind speed fluctuations, we calculate spectra from the simulation output of d03. For that, spectra are first calculated along the model grid rows (oriented approximately in west-east direction) and the model grid columns

(oriented approximately in south-north direction) (Müller et al., 2024). These spectra are then averaged over grid rows and columns.

*Author contributions.* SM, XGL designed the study with input from HF; HF provided and sorted the measurement data, SM performed the analysis and wrote the paper manuscript with input from XGL; XGL, HF, SM revised and edited the manuscript.

*Competing interests.* The authors declare that no competing interests are present.

*Acknowledgements.* The study is supported by the SDC project 906421. The authors acknowledge the computational and data resources provided by the Sophia HPC cluster at the Technical University of Denmark (DTU), DOI: 10.57940/FAFC-6M81. We thank Jacob Mann, Abdul Haseeb Syed, and Ásta Hannesdóttir for the valuable discussions.



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
