# Peer review of "Can the Mann model describe the typhoon turbulence?"

_Wind Energy Science, 2025_

## Referee Comment (RC1)

**Can the Mann model describe the typhoon turbulence?**

Reviewer's comments

**1 General comment**

The manuscript "Can the Mann model describe typhoon turbulence?" by Müller et al. examines the applicability of the uniform shear model (Mann, 1994), also known as Mann's model. The study addresses an important topic in wind energy science that deserves attention. While the paper is of broad international interest and falls within the scope of Wind Energy Science (WES), it could benefit from a clearer explanation of its relevance to the field. The analysis and methodology are rigorous, though some aspects may require improvement. The conclusions are generally well-supported by the results, except, perhaps, for the case of Typhoon Nuri in the back eyewall region (Figure 10 in the paper). I recommend a major revision. I am confident that the authors will be able to address my feedback, comments, and questions adequately. Below, I outline a few key points that I believe deserve further attention.

1. The study does not clearly discuss atmospheric stability, which is an important aspect of typhoon winds. It is likely that the stability in the four case studies is near-neutral or slightly convective. The shape of the turbulence spectra can change susbtantially as soon as the atmosphere become slightly convective. Since the uniform shear (US) model (Mann, 1994) is designed specifically for neutral atmospheric conditions, it may not adequately capture turbulence eddies generated by buoyancy forces. The unexpectedly high energy observed in the low-frequency range may be attributed to turbulence generated by convection rather than mesoscale fluctuations, although both processes could be found jointly. Addressing these aspects will provide a more complete characterization of the turbulence conditions in typhoon winds. As you may already know, the atmospheric stability can be estimated using the Obukhov length via the eddy covariance technique, which requires sonic temperature measurements and all three velocity components from 3D ultrasonic anemometers.

2. The scientific literature indicates that extreme wind loading leading to wind turbine structural collapse is often associated with non-stationary and non-Gaussian typhoon winds, particularly rapid changes in wind direction and speed. The present study focuses on stationary turbulence, which is a reasonable approach but also a choice that limit the scope of the study. Non-stationary typhoon winds have been explored in wind engineering literature, particularly since the 2000s. To better position the study within the wind energy and wind engineering it may be helpful to clearly state and discuss the place of the present study within the framework of stationary, Gaussian, homogeneous turbulence, from which the Mann model is based on.

3. It would be valuable to include an analysis of the skewness and kurtosis of velocity fluctuations to address the following question: Are (stationary) typhoon winds Gaussian? The study by Cao et al. (2009) suggests that the answer is "yes," but it would be valuable to see what findings emerge from the present study. This question is particularly relevant for wind turbine design, as non-Gaussian winds can lead to larger extreme wind loadings compared to Gaussian winds—a key assumption widely used in IEC standards and elsewhere. Such an analysis would be within the scope of the present study.

4. The study presents four key findings. However, finding number 4 appears to have been previously documented in Peña et al. (2010) or de Maré and Mann (2016) for non-typhoon winds, so its inclusion may not add significant new insights. Additionally, finding number 2 may require some revision, particularly in relation to atmospheric stability.

5. The study uses some dichotomous expressions that may be perceived as mutually exclusive by researchers in wind engineering and boundary-layer meteorology—fields that form the theoretical foundation for turbulent loading on wind turbines. To avoid ambiguity and potential misunderstandings, it may be helpful to elaborate on the use of these terms in both mesoscale and microscale meteorology. For example, the term "mesoscale turbulence" is used, but turbulence is often defined as wind fluctuations occurring at scales smaller than the mesoscale. To enhance clarity, a possible approach would be to use the term "mesoscale fluctuations" instead, as by Högström et al. (2002). A brief discussion of terminology could help ensure consistency and improve the study's readability for a broader audience.

6. In the main results, particularly in Figures 8–10, the vertical spectrum exhibits unusual behaviour in the inertial subrange. In this range, the ratios $F_w/F_u$ and $F_v/F_u$ would typically be expected to converge toward 1.33 under the assumption of local isotropy, or at the very least, remain above 1.2. However, in several measurements, this is not the case, and for Typhoons NURI and HAGUPIT, $F_w/F_u$ is even observed to be less than 1. This suggests the presence of significant flow distortion, the well-documented "w-bug," mast shadowing effects, or a combination of these factors. In contrast, for Typhoon PRAPIROON, the ratio $F_w/F_u$ appears to be close to 1.3 in the inertial subrange (Figure 9), which is an encouraging result. Also, the positive spectral slope observed in the inertial subrange for NURI (Back Inner and Eyewall) is indicative of an unphysical signal. Further investigation of these cases is needed, followed by a reassessment of the US model fit after conducting an in-depth quality check. Addressing these issues could potentially impact some of the study's key findings, including disregarding some of the data from typhoon NURI, which seems of lower quality than the other masts.

**2 Specific comments**

**Point 1**

**Introduction**: I think that a more specific and direct link to the design of wind turbines in typhoon-prone regions could strengthen the introduction. This would help highlight the relevance of the

topic. One possible way to enhance this aspect is by referencing recent events, such as the collapse of multiple wind turbines during Typhoon Yagi (Sanderson, 2024).

**Point 2**

**Introduction**: The literature review appears somewhat incomplete. The manuscript seems to align well with previous studies advocating for modifications to the IEC standard to account for extreme wind loading from typhoons (Chen and Xu, 2016). Also, it appears to complement nicely the findings of Cao et al. (2009), which suggest that the turbulence characteristics of typhoon winds closely resemble those of non-typhoon winds. There may be other studies of interest. Previous studies on typhoon winds for wind turbine design have not focused extensively on the Mann turbulence model, but Han et al. (2014) mention it briefly in their study. There are probably many more studies on Typhoon wind for wind loading on structures (bridges and tower). Maybe a summary table can be used? For example:

Table 1: Summary of studies on typhoon winds and their characteristics

| Study | Stationary/Non-stationary | Gaussian/Non-Gaussian | Structure Type | Turbulence Model |
|---|---|---|---|---|
| Present study | Stationary | — | Wind Turbine | Mann |
| author et al (year) | Non-stationary | Non-Gaussian | Bridge | - |
| author et al (year) | Stationary | Gaussian | Wind Turbine | Mann Model |
| author et al (year) | Non-stationary | Non-Gaussian | Tower | Kaimal Spectrum |
| author et al (year) | Non-stationary | Non-Gaussian | Bridge | ESDU Model |
| author et al (year) | Stationary | Gaussian | Wind Turbine | IEC Kaimal Model |

**Point 3**

Page 2, lines 33–34: The interpretation of sea spray in this context seems somewhat unusual. While it is reasonable to mention it, a positive slope in the inertial subrange of the normalized spectra is, in my experience, typically indicative of noise. This can arise from various sources, such as rain causing artificially high velocity readings in sonic anemometer data, tower shadowing effects, or aliasing. The correct reference might be Li et al. (2015) rather than Li et al. (2012). Overall, the unusual behavior observed in the inertial subrange may not be physical but rather a reflection of instrumental errors. Sonic anemometers are known to perform poorly under heavy rain or when exposed to water spray, which could explain this anomaly.

**Point 4**

Page 2, line 47: The Kaimal model used in the IEC standard differs significantly from the original model by Kaimal et al. (1972)—so much so that referring to it simply as Kaimal may be misleading. It might be more appropriate to refer to it as IEC-Kaimal to distinguish it from the original formulation. Alternatively, citing the IEC standard directly, rather than the original paper by Kaimal

et al. (1972), could provide better clarity.

**Point 5**

Page 2, line 52: Stating that the Mann model fails to account for mesoscale fluctuations may not be accurate, as it does not attempt to model them in the first place. A more precise phrasing would acknowledge that the Mann model is designed specifically for turbulence and does not incorporate mesoscale fluctuations by definition.

**Point 6**

Page 2, lines 54–55: The reference to inactive turbulence in Högström et al. (2002) could be misleading. If I remember properly, the authors actually argue against using this term. What they describe as inactive turbulence still falls within the definition of turbulence and should not be conflated with mesoscale motion. It may be useful to clarify this distinction to avoid potential misinterpretations.

**Point 7**

Pages 2–3, lines 53–63: The terminology used in this paragraph appears to conflate mesoscale fluctuations with turbulence, which may lead to conceptual ambiguities. The distinction between mesoscale and turbulent motions is well-established in atmospheric science. For instance, Högström et al. (2002) describe mesoscale fluctuations as "unsteady quasi-two-dimensional motion," emphasizing that they are non-turbulent. Typically, mesoscale fluctuations lie on the left side of the spectral gap, while turbulence is on the right. The spectral gap, which separates these two scales, is a key feature of atmospheric turbulence spectra. Under convective conditions, this gap may become less distinct or even undetectable due to buoyancy-generated turbulence overlapping with mesoscale motions. However, referring to these large-scale fluctuations as mesoscale turbulence may be misleading. It would be beneficial to clarify this distinction to ensure the terminology aligns with established turbulence theory. Specifically, rather than mesoscale turbulence, a more precise term might be mesoscale fluctuations or mesocale motion.

**Point 8**

Table 1: Many Gill WindMaster Pro anemometers produced between 2006 and 2015 were affected by a known issue that led to an underestimation of the vertical wind component. See, for example, `https://www.licor.com/support/EddyPro/topics/w-boost-correction.html`. Would it be possible to verify whether this issue affected the instruments used in this study? If so, the bias can be corrected (to some extent) using a straightforward data processing method, as described in the linked resource.

**Point 9**

Line 81: Could a brief explanation be provided for the choice of the 6 km area? If this selection is related to internal boundary layers, would it be possible to use a simple analytical model to estimate the internal boundary layer thickness? Garratt (1990) presents several relevant models that might be useful. If such an approach is considered, specifying the roughness length for the sectors of interest would further clarify the reasoning behind the choice.

**Point 10**

**Spike Filtering**: It may be beneficial to first apply a flat threshold, such as 65 m/s, which is the upper measurement limit of the Gill sonic anemometer. The reason for this is that spikes can exceed this value, potentially masking other outliers. A possible approach could be: (1) Apply a flat threshold to remove physically unrealistic values. (2) Perform outlier detection using a moving median filter. It is important to use the absolute median deviation (MAD) rather than the absolute mean deviation, as recommended by Leys et al. (2013). It is currently unclear whether the study employs the median or arithmetic mean for outlier detection. The wording "average" suggests the latter, but clarification would be helpful.

**Point 11**

**Data Processing and Data Filling**: The use of linear interpolation for datasets with 15% missing values (NaNs) raises some concerns, particularly for turbulence studies, where preserving statistical properties is crucial. In atmospheric science, a common threshold for acceptable missing data is around 5%. How would the findings be affected if a stricter threshold were applied, such as 10% or 5% NaNs? Exploring the sensitivity of the results to different thresholds could help assess the robustness of the analysis.

**Point 12**

Section 2.1.2 – Power Spectra Calculation: The approach described in this section closely resembles Welch's method (Welch, 2003), which is a well-established technique for power spectral estimation. To avoid unnecessary reinvention, it may be beneficial to explicitly state that Welch's method is being used and to reference the appropriate implementation, such as the scipy.signal.welch function in Python or the pwelch function in MATLAB. In this study suggest using around 3 segments with 50% overlapping to reduce uncertainties. Reformulating this section to reflect this could improve clarity and align the methodology with standard signal processing practices..

**Point 13**

Section 2.1.2 – Stationarity Test: Using wind direction change as a criterion for stationarity is a good idea. However, if the goal is to analyze stationary fluctuations specifically, it may also be useful to check stationarity in mean wind speed (first-order stationarity) and variance (second-order stationarity). While this might not be strictly necessary given that the results appear reasonable,

performing these additional tests could provide a more rigorous assessment for future studies.

**Point 14**

Line 154–155: A clearer formulation might be to state that a key advantage of the Mann model is that the second-order structure of homogeneous atmospheric turbulence in a neutral atmosphere is incorporated using only three parameters. The limited number of parameters is a significant advantage, particularly for wind energy applications, where simplicity and computational efficiency are crucial.

**Point 15**

Line 180–181: The authors raise an important point: the study focuses on stationary turbulence. This distinction should be explicitly mentioned in the abstract, as many readers might initially expect the paper to address non-stationary turbulence for Typhoon winds.

**Point 16**

Line 180–181 and lines 224: The claim that significant wind direction changes make the Fourier transform "invalid" is not accurate in my opinion. The Fourier transform remains valid for both stationary and non-stationary signals because it preserves phase information. However, the power spectral density (PSD), which discards phase information, is primarily suited for stationary signals. In strongly non-stationary conditions, such as in the eye and rainband regions of a tropical cyclone, the PSD may not accurately represent the underlying turbulence characteristics. A more precise formulation could be to state that the spectral analysis is not performed in the eye and rainband regions due to the strong non-stationarity of the wind field, which complicates the interpretation of the power spectra.

**Point 17**

Figures 4-7: The figures are well-presented. However, why are the Mann parameters shown only in Figures 4-5 and not in Figures 6-7? If there is a reason for this omission, it would be helpful to clarify it.

**Point 18**

Figures 4-5: As noted by the authors, some fits have reached the upper boundary for $\Gamma$ and $L$. This suggests that the fitting procedure may not have converged. While such cases can occur, it might be beneficial to disregard those samples (if this has not already been done) or explore whether a better initial guess could improve the fitting.

**Point 19**

**Mast shadowing:** During the passage of a typhoon, wind direction can change dramatically, which may lead to periods where the sonic anemometer data are affected by mast shadowing. To ensure a meaningful analysis, it would be useful to provide the boom orientation and indicate whether and when the data are impacted by mast shadowing. If data are found to be affected, they should be excluded from the analysis.

**Point 20**

Lines 267-268: This is a good observation. As noted in the manuscript, the length scale parameter ($L$) tends to increase with height above the surface. This trend has been documented in previous studies. Also, it is generally expected that $L$ will increase substantially under convective atmospheric conditions, as discussed in Sathe et al. (2013).

**Point 21**

Lines 272-273: The variation of $\alpha\varepsilon^{2/3}$ is influenced not only by the mean wind speed but also by the variance of the velocity components. The current statement suggests a dependence solely on the mean wind speed, but it would be more accurate to acknowledge both factors.

**Point 22**

Figure 10: Could the secondary peak at $k > 1 \times 10^{-1}$ m$^{-1}$ be related to the first eigenfrequency of the mast? If so, it may be useful to investigate whether structural resonance effects influence the spectral shape.

**Point 23**

Line 355: Good observation. It is possible that many turbulence models used for wind loading underrepresent low-wavenumber variability in the $v$-component. However, I ignore what could be the physical interpretation of this variability. Boundary layer rolls, as proposed in this study seems to be a reasonable interpretation.

**Point 24**

Line 342-344: This statement could be rephrased with more caution. Some of the observed spectral peaks do appear to have a physical origin—for instance, the secondary peak in the $F_v$-spectrum for the Back Outer region is consistent with trends previously observed in the literature. However, other peaks, such as those for the Front Outer (FO) region, may be influenced by large uncertainties in the lowest-frequency bins due to the use of the modified periodogram method.

It is important to remember that power spectral densities (PSDs) are statistical estimates and inherently contain uncertainties that increase as the wavenumber decreases. The spectral peaks observed at $k < 3 \times 10^{-3}$ m$^{-1}$ may be particularly affected by these uncertainties. Ideally, a

larger sample size would help determine whether these peaks have a physical basis, but obtaining additional typhoon wind data is understandably challenging. An alternative approach could be to use a greater number of overlapping segments (with at least 50% overlap) in the Welch algorithm to reduce spectral uncertainties and smooth out noise in the lowest frequencies.

**Point 25**

**Spectrum shape**: Hagupit's spectrum shows a flattened spectral peak, which raises the question of whether ground blockage effects contribute to this flattening. A potential way to assess the impact of ground blockage is to analyze the imaginary part of the cross-spectrum between $u$ and $w$, as suggested in Mann (1994).

**Point 26**

Lines 346-348: The presence of nonzero covariance $\overline{v'w'}$ in offshore conditions has been documented since the 1980s (see Geernaert (1988)), so this phenomenon is now well established. Maybe these observations should not be described as non-classical turbulence behaviour. Given that the measurements in this study are from coastal or offshore regions, observing significant values in the $F_{vw}$ co-spectrum is expected. It may be useful to mention this as a reminder for readers interested in offshore wind turbine design.

**Point 27**

Lines 350-351: This sentence may need to be rephrased after conducting a more in-depth literature review. While there have been numerous relevant studies, many of them belong to the field of wind engineering, which, somewhat surprisingly, is often overlooked by the wind energy community. This study presents an opportunity to highlight the overlap between these two fields and possibly propose a path toward a unified description of turbulence in tropical cyclones for wind loading on structures.

**Point 28**

Section 4, paragraph 1: The first paragraph of this section reads more like a conclusion. It could either be removed or merged into Section 5 (Conclusion).

**Point 29**

Line 359; The Mann model has, to my knowledge, been tested against actual measurements in typhoon winds (see Han et al. (2014)), but only in a superficial manner. In contrast, this study provides a much more detailed analysis, making it both valuable and complementary to previous work.

**Point 30**

Discussion section: It may be useful to discuss how this study fits within the broader framework of stationary vs. non-stationary typhoon wind models. The absence of the Mann model in studies of non-stationary turbulence is reasonable, as it was not designed for such conditions. However, most wind-induced damage to wind turbines during typhoons is likely caused by extreme, non-stationary gusts. Future studies might explore ways to combine the Mann model with non-stationary models, for example, by incorporating a fluctuating mean wind speed and direction. This could be briefly outlined in the discussion as a potential avenue for further research.

**Point 31**

Table 3: This is an informative table, but the comparison of spectral slopes may be misleading, as some slopes in the inertial subrange appear to be spurious. This could mislead readers and practitioners into developing inappropriate wind loading models. Instead of presenting the slope as a result, it may be more appropriate to treat it as a quality check for the data. The last column, which presents the logarithm of spectral peaks, is particularly interesting. It could be beneficial to expand on this aspect.

**Point 32**

The term "excessive energy" may need to be changed into something like "energy larger than predicted by turbulence models designed for a neutral atmosphere". The energy in the $u$ and $v$ components is only excessive if it results from an error. Offshore measurements consistently show that these components exhibit higher energy at lower wavenumbers than predicted by the Kaimal model or Mann model. This effect appears to be even more pronounced for typhoon winds, though it may also be influenced by convective conditions. The literature documents a substantial increase in energy of the low-frequency $F_u$ and $F_v$ spectra when the atmosphere becomes slightly convective. This suggests that the observed energy enhancement in the low-wavenumber range is more likely linked to buoyancy effects rather than purely mesoscale fluctuations.

**Point 33**

Line 374-376: The interpretation that the increased energy at lower wavenumbers is due to mesoscale fluctuations may be criticized. The time and spatial scales used in this study—30-minute averaging with PSD estimation using overlapping segments (potentially reducing it to 10-minute windows)—indicate that the observed structures still fall within the turbulence regime. In fact, in multiple instances within this study, the spectral gap is only just becoming visible, reinforcing the idea that the large eddies observed here are still turbulence—though likely strongly anisotropic or influenced by buoyancy effects. Also, the atmosphere might be slightly convective, which would amplify turbulence through buoyancy-generated eddies, an effect not accounted for in the Mann model. Since the Mann model was developed specifically for neutral conditions, deviations from the model in the low-frequency range under stable or unstable stratification are expected.

In cases where the spectral gap is not visible within a 30 min window (e.g. NURI, Front Outer), the observed spectral behavior is more likely to be convective turbulence than mesoscale fluctuations. Under convective conditions, large turbulence eddies generated by buoyancy can fill the spectral gap, creating a transition zone where turbulence and mesoscale fluctuations may overlap. However, identifying this overlap reliably requires longer averaging times—likely closer to 1 hour or more rather than the 30-minute windows used in this study. If the spectral gap remains invisible even with 1+ hour of data, the observed variability may represent a mixture of large turbulence eddies and mesoscale fluctuations. In such cases, distinguishing between the two might require a turbulence model that explicitly accounts for unstable atmospheric stratification.

**Point 34**

Lines 432–440: The statements in this section may be controversial. Aeroelastic codes do not necessarily require spatial series. Since the dynamic response is computed in the time domain by solving the equation of motion, the required input consists of time series, not spatial series. Additionally, this paragraph seems to paraphrase the assumption of stationarity, and I am unsure if it adds substantial value to the paper. A more straightforward approach would be to state that if the assumption of stationarity holds, the methods used in IEC standards can be applied. If it does not hold, a different framework is required. More generally, IEC standards are based on the assumption of stationary, homogeneous, Gaussian, and ergodic turbulence. However, when this assumption is violated—as is often the case for typhoon winds—a different framework must be adopted. This alternative framework is commonly referred to as non-stationary turbulence. Furthermore, non-stationary turbulence frequently exhibits non-Gaussian and non-homogeneous characteristics, as turbulence that deviates from stationarity often displays non-Gaussian properties. This is particularly relevant because non-Gaussian wind loading is known to produce much higher extreme loads than Gaussian wind loading (Gong and Chen, 2014). To my knowledge, this alternative framework has not yet been clearly established in wind energy research. However, it has been extensively studied in wind engineering for bridge design and high-rise building design since the early 2000s.

**Point 35**

Lines 447–451: The critical role of coherence in wind loading has been well recognized since the 1960s (see Davenport (1961)). Therefore, citing additional foundational studies rather than relying solely on Dimitrov et al. as a reference may be a good idea. From the results in Mann (1994) and Cheynet (2019), it appears that Mann's model tends to overestimate the coherence of the $u$-component at vertical separations. This behavior may also hold for stationary typhoon wind conditions, but further investigation would be needed to confirm this. Previous studies indicate that Mann's model may provide reasonable estimates of turbulence coherence at short lateral separations, but there is a knowledge gap for the case of large lateral separations.

**Point 36**

**Conclusions:** The conclusion may need to be reformulated and be more closely related to wind energy science.

For points 1, it could be a good idea to further elaborate on two key aspects. First, there are large uncertainties at low wavenumbers in the spectra, which should be acknowledged more explicitly. Second, the possible unstable stratification of the atmosphere implies that turbulence eddies generated by buoyancy may be contributing to the observed spectral features. This is not accounted for in the Mann model, as it was developed specifically for neutral atmospheric conditions. The approach using mesoscale simulations could help assess to what extent mesoscale fluctuations and turbulence interact within a 30-minute window, which could be an interesting topic for future research.

Maybe point 2 can be removed from the findings. Instead, the following points could be added: (i) There are regions within a typhoon where the wind can be described as stationary. In these cases, the Mann model provides a reasonable approximation of the turbulence structure, which is a valuable result. (ii) The spectral peaks of the *u*- and *v*-spectra occur at much closer frequencies in this study than predicted by the Mann model. This raises the question of whether this discrepancy is related to a non-neutral atmospheric state or typhoon wind characteristics. This could be explored further.

Point 4 might be removed from the key findings. Similar findings have already been reported in non-typhoon conditions. Since this behavior is already well documented, it may not be considered as a "new finding". For point 5, the behavior described has been well known since the 1980s for offshore wind and is expected to apply to typhoon winds in coastal regions as well. Given its well-documented nature, it may not need to be emphasized as a key new finding.

**3 Technical comments**

**Point 1**

On the use of $10^{-2.5}$ m$^{-1}$: writing $3 \times 10^{-3}$ m$^{-1}$ is easier to read than $10^{-2.5}$ m$^{-1}$.

**Point 2**

Spelling convention: Since Copernicus Publications is a European publisher, it may be preferable to use British English rather than American English for consistency with their style.

**Point 3**

Page 1, line 9 – Use of "excessive energy": The term "excessive energy" may not be the most appropriate choice, as the model underestimates the observations. A more precise phrasing, such as "more energy than predicted," could better reflect the context.

**Point 4**

Section 3.3: The title of the subsection may be renamed or adjusted using an question mark "?"

**Point 5**

The units of the spectra $kF_i$, where $i = u, v, w$, should be either in m s$^{-1}$ or m s$^{-2}$ Hz$^{-1}$, given that $F$ is expressed in m$^2$ s$^{-1}$ and $k$ is in m$^{-1}$. However, Table 3 (and possibly other sections) lists the units as m$^2$ s$^{-2}$, which appears inconsistent. It would be useful to verify and correct the units where necessary.

**References**

Mann, J.. The spatial structure of neutral atmospheric surface-layer turbulence. Journal of fluid mechanics 1994;273:141–168.

Cao, S., Tamura, Y., Kikuchi, N., Saito, M., Nakayama, I., Matsuzaki, Y.. Wind characteristics of a strong typhoon. Journal of wind engineering and industrial aerodynamics 2009;97(1):11–21.

Peña, A., Gryning, S.E., Mann, J., Hasager, C.B.. Length scales of the neutral wind profile over homogeneous terrain. Journal of Applied Meteorology and Climatology 2010;49(4):792–806.

de Maré, M., Mann, J.. On the space-time structure of sheared turbulence. Boundary-Layer Meteorology 2016;160(3):453–474.

Cheynet, E.. Influence of the measurement height on the vertical coherence of natural wind. In: Proceedings of the XV Conference of the Italian Association for Wind Engineering: IN-VENTO 2018 25. Springer; 2019, p. 207–221.

Högström, U., Hunt, J., Smedman, A.S.. Theory and measurements for turbulence spectra and variances in the atmospheric neutral surface layer. Boundary-Layer Meteorology 2002;103:101–124.

Sanderson, C.. Why were 'typhoon-proof' chinese wind turbines flattened by typhoon yagi? 2024. URL: https://www.rechargenews.com/wind/why-were-typhoon-proof-chinese-wind-turbines-flattened-by-typhoon-yagi-/2-1-1707314; accessed: 2025-02-17.

Chen, X., Xu, J.Z.. Structural failure analysis of wind turbines impacted by super typhoon usagi. Engineering failure analysis 2016;60:391–404.

Han, T., McCann, G., Mücke, T., Freudenreich, K.. How can a wind turbine survive in tropical cyclone? Renewable energy 2014;70:3–10.

Garratt, J.. The internal boundary layer—a review. Boundary-layer meteorology 1990;50:171–203.

Leys, C., Ley, C., Klein, O., Bernard, P., Licata, L.. Detecting outliers: Do not use standard deviation around the mean, use absolute deviation around the median. Journal of experimental social psychology 2013;49(4):764–766.

Welch, P.. The use of fast Fourier transform for the estimation of power spectra: A method based on time averaging over short, modified periodograms. IEEE Transactions on audio and electroacoustics 2003;15(2):70–73.

Sathe, A., Mann, J., Barlas, T., Bierbooms, W., Van Bussel, G.. Influence of atmospheric stability on wind turbine loads. Wind Energy 2013;16(7):1013–1032.

Geernaert, G.. Measurements of the angle between the wind vector and wind stress vector in the surface layer over the North Sea. Journal of Geophysical Research: Oceans 1988;93(C7):8215–8220.

Gong, K., Chen, X.. Influence of non-gaussian wind characteristics on wind turbine extreme response. Engineering structures 2014;59:727–744.

Davenport, A.G.. The spectrum of horizontal gustiness near the ground in high winds. Quarterly Journal of the Royal Meteorological Society 1961;87(372):194–211.

---

## Referee Comment (RC2)

Review comments to the manuscript WES-2025-7 "Can the Mann model describe the typhoon turbulence?"

This study tries to investigate the turbulence characteristics during tropical cyclones by analyzing the measurement data. The analysis of the measurement data is detailed and there seems no problems in the analysis. However, there is a big jump in logic between the facts that the authors show and the conclusions. The authors conclude that the gap between the measurement data and Mann model is caused by the characteristics of tropical cyclones but the authors does not show enough facts to justify this conclusion. The authors should show the difference of the spectrum from measurement data during tropical cyclone and other strong wind causing events to discuss the differences between the tropical cyclones and other strong wind causing events (e.g., extratropical cyclones), If not, the discussion in this paper seems almost meaningless and it is difficult to accept this manuscript for publication. The reviewer would suggest the authors to rewrite the manuscript considering the following comments.

Followings are more specific questions. Some of them are related to the point mentioned above and the other comments are smaller points.

- 1. Line 110. Is it typical that de-spiking is needed for sonic anemometer measurement? Is it only for extreme wind events? Anyway, in this case the authors should show some examples of de-spiking by showing time history of the original data and filtered data to justify their filtering.
- 2. Line 135. How was the value of U in the Taylor hypothesis calculated? Is it the average wind speed for each 30 minutes?? However, can it be justified for the case when the wind speed changes a lot, which is typical when tropical cyclone is approaching?
- 3. Fig 4-7. What is the source of the paths of the typhoon? I would like authors to provide the source of these paths.
- 4. Fig 4-7. The discussion on the low wave number side of the spectrum is little unclear. If It seems that some of the spectrum (u or v component) shows fairy good agreement with Mann model, and it would be difficult to say that Mann model does not work for tropical cyclones. Even at the inner part (e.g., front inner of Hagpit), the fit to Mann model is not bad. If the authors would like to say that these low wave number characteristics are

- specific to tropical cyclones, then the comparison with non-tropical cyclone events are needed, which is missing in the current manuscript.
- 5. Line 388- The connection between the measurement data analysis in section 3 and the analysis of WRF simulation is little unclear. The wave number range in Figure 11 and Figures 4-7 are different and the reviewer have difficulty to understand the connection between them. More clarifications are needed.
- 6. Figure 11, What are the cause of the "roll"? Obviously, it cannot be concluded from the facts provided in this paper that this is specific to tropical cyclones. As is clear from figure 11 (a), the roll is visible on the left side of the tropical cyclone, where wind is northerly, and maybe the land affects the forming of this roll. On the other hand, on the right side of the tropical cyclone, the roll is not visible, maybe because of the southerly wind from the sea. But if so, this phenomenon has little connection with tropical cyclones and similar phenomena could also be visible in the other strong wind events. If authors would like to point out that the roll and associated large power in low wave number regions, the comparison with non-tropical cyclone events are needed.
- 7. Figure 11 and figure 5. The centre of the tropical cyclone seems to be at 22 degrees N and 115 degrees E at 0300UTC on August 22nd in figure 11. However, in figure 5, the path does not pass this point. Is it due to the error in the WRF simulation? or is it the error in the tropical cyclone path analysis? Anyway, comments from authors are needed.

---

## Author Comment (AC1)

**Answer to reviewer comments of referee 1**

January 28, 2026

Dear reviewer, we would like to thank you for thorough and constructive feedback on our manuscript. We highly appreciate your input. Please find our answer in black following your comments in green.

**1 General comments**

The manuscript "Can the Mann model describe typhoon turbulence?" by Müller et al. examines the applicability of the uniform shear model (Mann, 1994), also known as Mann's model. The study addresses an important topic in wind energy science that deserves attention. While the paper is of broad international interest and falls within the scope of Wind Energy Science (WES), it could benefit from a clearer explanation of its relevance to the field.

Thank you for this comment. The introduction has been updated as summarized in point 1 of the Specific comments.

The analysis and metodology are rigorous, though some aspects may require improvement. The conclusions are generally well-supported by the results, except, perhaps, for the case of Typhoon Nuri in the back eyewall region (Figure 10 in the paper).

Thank you for this comment. The concern about the quality of the measurements underlying the large spectral energy at high frequencies in Figure 10 is valid. We addressed this issue by explicitly stating in several sections of the manuscript, including the abstract, the caption of Table 3, as well as the Results, Discussion, and Conclusions, that this is likely related to measurement artifacts. Nevertheless, we chose to keep the figure and associated discussion on the excessive spectral energy at large wavenumbers in the manuscript for several reasons. First, doing so allows us to continue the discussion initiated by Li et al. (2015), and taken up in further studies such as Tao and Wang (2019). Second, although it is likely, we cannot conclude with certainty that the enhanced spectral energy is caused by tower vibrations, tower shadowing, or artificial spikes. Clearly, more research and measurements are needed to further advance our understanding.

I recommend a major revision. I am confident that the authors will be able to address my feedback, comments, and questions adequately. Below, I outline a few key points that I believe deserve further attention.

Thank you very much for the detailed, constructive review!

1. The study does not clearly discuss atmospheric stability, which is an important aspect of typhoon winds. It is likely that the stability in the four case studies is near-neutral or slightly convective. The shape of the turbulence spectra can change susbtantially as soon as the atmosphere become slightly convective. Since the uniform shear (US) model (Mann, 1994) is designed specifically for neutral atmospheric conditions, it may not adequately capture turbulence eddies generated by buoyancy forces. The unexpectedly high energy observed in the low-frequency range may be attributed to turbulence generated by convection rather than mesoscale fluctuations, although both processes could be found jointly. Addressing these aspects will provide a more complete characterization of the turbulence conditions in typhoon winds. As you may already know, the atmospheric stability can be estimated using the Obukhov length via the eddy covariance technique, which requires sonic temperature measurements and all three velocity components from 3D ultrasonic anemometers.

   Thank you for pointing this out. Following your suggestion, we assessed atmospheric stability by estimating the kinematic heat flux from the ultrasonic anemometer measurements at 60 m and computing the corresponding Obukhov length. The results are in the same range as reported in Li et al. (2015) for typhoons Hagpuit and Nuri. Our analysis shows no evidence of convective conditions in the outer regions of the cyclones, where the more energy was found in the typhoon spectra, than predicted by the Mann model. Overall, the kinematic heat fluxes were small or even overall weakly negative, suggesting that the conditions were not convective. The only exception is typhoon Nuri, where positive kinematic heat fluxes were found in the back eyewall and back inner region. To improve transparency, we have included figures showing the calculated heat flux, friction velocity, and Obukhov length for each of the four typhoons in Sect. 4 of this document.

2. The scientific literature indicates that extreme wind loading leading to wind turbine structural collapse is often associated with non-stationary and non-Gaussian typhoon winds, particularly rapid changes in wind direction and speed. The present study focuses on stationary turbulence, which is a reasonable approach but also a choice that limit the scope of the study. Non-stationary typhoon winds have been explored in wind engineering literature, particularly since the 2000s. To better position the study within the wind energy and wind engineering it may be helpful to clearly state and discuss the place of the present study within the framework of stationary, Gaussian, homogeneous turbulence, from which the Mann model is based on.

   We appreciate this comment. Wind conditions during tropical cyclones are indeed inherently non-stationary. However, our goal was not to restrict the analysis to stationary wind conditions. In fact, we explicitly aimed to capture the spectral energy at timescales between approximately 10 and 30 minutes related to non-stationary features. Specifically, we wanted to show that the Mann model, a microscale spectral model, cannot represent these features. This is now clearly stated in the Data and Methods section 2.2.1. Based on your feedback, we have placed the study in a better context of existing literature employing methods tailored to non-stationary wind characteristics. In the introduction, we have included relevant literature addressing non-stationary frameworks to analyze tropical cyclone wind fields from the bridge engineering sector, and in the discussion, we suggest integrating that body of literature into the wind energy sector as well.

3. It would be valuable to include an analysis of the skewness and kurtosis of velocity fluctuations to address the following question: Are (stationary) typhoon winds Gaussian? The study by Cao et al. (2009) suggests that the answer is "yes," but it would be valuable to see what findings emerge from the present study. This question is particularly relevant for wind turbine design, as non-Gaussian winds can lead to larger extreme wind loadings compared to Gaussian winds—a key assumption widely used in IEC standards and elsewhere. Such an analysis would be within the scope of the present study.

   This is an interesting point. Following your comment, we examined the skewness and kurtosis. Similar to Cao et al. (2009), we found that the turbulence was nearly Gaussian, with a mean skewness of 0.0 and a kurtosis of 2.9. We have included this finding in the discussion.

4. The study presents four key findings. However, finding number 4 appears to have been previously documented in Peña et al. (2010) or de Maré and Mann (2016) for non-typhoon winds, so its inclusion may not add significant new insights.

   You are right, previous studies have shown that the Mann model overestimates the $uw$-cospectra at the Danish sites Høvsøre and Østerild. We have reformulated this point to highlight the similarity to non-typhoon cases: "*Further deviations of the Mann model in the uw- and vw-cospectra have previously been reported under non–tropical cyclone conditions. These deviations are also evident in the analyzed typhoon cases: the Mann model overestimates energy in the uw-cospectra, consistent with findings from different sites in Denmark (Peña et al., 2010; Peña, 2019).*" Additionally, finding number 2 may require some revision, particularly in relation to atmospheric stability.

   We understand your concern about the quality of the measurements underlying finding number 2. We now acknowledge that this finding may be related to instrument noise, tower shadowing, or tower vibrations. Regarding atmospheric stability, we assume the concern mainly relates finding number 1, which is addressed in the general comment number 1.

5. The study uses some dichotomous expressions that may be perceived as mutually exclusive by researchers in wind engineering and boundary-layer meteorology—fields that form the theoretical foundation for turbulent loading on wind turbines. To avoid ambiguity and potential misunderstandings, it may be helpful to elaborate on the use of these terms in both mesoscale and microscale meteorology. For example, the term "mesoscale turbulence" is used, but turbulence is often defined as wind fluctuations occurring at scales smaller than the mesoscale. To enhance clarity, a possible approach would be to use the term "mesoscale fluctuations" instead, as by Högström et al. (2002). A brief discussion of terminology could help ensure consistency and improve the study's readability for a broader audience.

   Thank you for the suggestion. We have adopted the term "mesoscale fluctuations." Additionally, we have improved the introduction to the term: "*Wind speed power spectra often exhibit a spectral gap that distinguishes three-dimensional, microscale turbulence from two-dimensional, larger-scale fluctuations. The latter has been referred to by different terms in the literature. In this study, the term "mesoscale fluctuations" is used.*"

6. In the main results, particularly in Figures 8–10, the vertical spectrum exhibits unusual behaviour in the inertial subrange. In this range, the ratios $F_w/F_u$ and $F_v/F_u$ would typically be expected to converge toward 1.33 under the assumption of local isotropy, or at the very least, remain above 1.2. However, in several measurements, this is

not the case, and for Typhoons NURI and HAGUPIT, $F_w/F_u$ is even observed to be less than 1. This suggests the presence of significant flow distortion, the well-documented "w-bug," mast shadowing effects, or a combination of these factors. In contrast, for Typhoon PRAPIROON, the ratio $F_w/F_u$ appears to be close to 1.3 in the inertial subrange (Figure 9), which is an encouraging result. Also, the positive spectral slope observed in the inertial subrange for NURI (Back Inner and Eyewall) is indicative of an unphysical signal. Further investigation of these cases is needed, followed by a reassessment of the US model fit after conducting an in-depth quality check. Addressing these issues could potentially impact some of the study's key findings, including disregarding some of the data from typhoon NURI, which seems of lower quality than the other masts.

Thank you for your comments and excellent suggestions regarding data filtering and the W-bug in the specific comments 8, 10, and 11. We have implemented your recommendations, corrected the W-bug, and rerun the data despiking. Indeed we found reason to exclude additional measurements, particularly from Nuri's eyewall region. However, the newly calculated spectra still exhibit similar behavior with respect to the discussed points. This increases our confidence in the robustness of the findings. Nevertheless, as you remark mast shadowing and tower vibrations cannot be ruled out as potential causes of the large spectral energy at small scales. This remark has been included in several locations, including the conclusions and the abstract. We note that more measurements are needed to improve our understanding.

**2  Specific comments**

1. Introduction: I think that a more specific and direct link to the design of wind turbines in typhoon-prone regions could strengthen the introduction. This would help highlight the relevance of the topic. One possible way to enhance this aspect is by referencing recent events, such as the collapse of multiple wind turbines during Typhoon Yagi (Sanderson, 2024).
   This is an excellent idea, we followed your suggestions and added three examples of recent damages to wind turbines during tropical cyclones.

2. Introduction: The literature review appears somewhat incomplete. The manuscript seems to align well with previous studies advocating for modifications to the IEC standard to account for extreme wind loading from typhoons (Chen and Xu, 2016). Also, it appears to complement nicely the findings of Cao et al. (2009), which suggest that the turbulence characteristics of typhoon winds closely resemble those of non-typhoon winds. There may be other studies of interest. Previous studies on typhoon winds for wind turbine design have not focused extensively on the Mann turbulence model, but Han et al. (2014) mention it briefly in their study. There are probably many more studies on Typhoon wind for wind loading on structures (bridges and tower). Maybe a summary table can be used? (example given). Thank you for providing these references; we have included them. In addition, we added several references from bridge engineering that specifically address nonstationarity in tropical cyclones (Xu and Chen (2004); Wang et al. (2018); Tao and Wang (2019)). Please note that we slightly revised the structure of the Introduction to incorporate the added literature on nonstationarity.

3. Page 2, lines 33–34: The interpretation of sea spray in this context seems somewhat unusual. While it is reasonable to mention it, a positive slope in the inertial subrange of the normalized spectra is, in my experience, typically indicative of noise. This can arise from various sources, such as rain causing artificially high velocity readings in sonic anemometer data, tower shadowing effects, or aliasing. The correct reference might be Li et al. (2015) rather than Li et al. (2012). Overall, the unusual behavior observed in the inertial subrange may not be physical but rather a reflection of instrumental errors. Sonic anemometers are known to perform poorly under heavy rain or when exposed to water spray, which could explain this anomaly. Thank you for pointing out the error in the citation. We agree, that unusual large spectral energy at high-frequencies could be caused by measurement artifacts. To make this clear we added: "However it could not be ruled out that it could also be caused by measurement artifacts such as tower wake effects, tower vibrations, or sensor noise (Barthlott and Fiedler, 2003; Gao et al., 2024; Kaimal and Finnigan, 1994)".

4. Page 2, line 47: The Kaimal model used in the IEC standard differs significantly from the original model by Kaimal et al. (1972)—so much so that referring to it simply as Kaimal may be misleading. It might be more appropriate to refer to it as IEC-Kaimal to distinguish it from the original formulation. Alternatively, citing the IEC standard directly, rather than the original paper by Kaimal et al. (1972), could provide better clarity. Good point, we changed the sentence to: "The IEC standard recommends using a modified version of the Kaimal model (Kaimal et al., 1972) or the Mann uniform shear model (Mann, 1994) [...]".

5. Page 2, line 52: Stating that the Mann model fails to account for mesoscale fluctuations may not be accurate, as it does not attempt to model them in the first place. A more precise phrasing would acknowledge that the Mann model is designed specifically for turbulence and does not incorporate mesoscale fluctuations by definition. That's right, we followed your suggestion.

6. Page 2, lines 54–55: The reference to inactive turbulence in Högström et al. (2002) could be misleading. If I remember properly, the authors actually argue against using this term. What they describe as inactive turbulence still falls within the definition of turbulence and should not be conflated with mesoscale motion. It may be useful to clarify this distinction to avoid potential misinterpretations. This is a fair point. We have reformulated the paragraph and decided to refrain from using the term inactive turbulence, also in connection with General comment 5 and Specific comment 7.

7. Pages 2–3, lines 53–63: The terminology used in this paragraph appears to conflate mesoscale fluctuations with turbulence, which may lead to conceptual ambiguities. The distinction between mesoscale and turbulent motions is well-established in atmospheric science. For instance, Högström et al. (2002) describe mesoscale fluctuations as "unsteady quasi-two-dimensional motion," emphasizing that they are non-turbulent. Typically, mesoscale fluctuations lie on the left side of the spectral gap, while turbulence is on the right. The spectral gap, which separates these two scales, is a key feature of atmospheric turbulence spectra. Under convective conditions, this gap may become less distinct or even undetectable due to buoyancy-generated turbulence overlapping with mesoscale motions. However, referring to these large-scale fluctuations as mesoscale turbulence may be misleading. It would be beneficial to clarify this distinction to ensure the terminology aligns with established turbulence theory. Specifically, rather than mesoscale turbulence, a more precise term might be mesoscale fluctuations or mesoscale motion. Thank you for pointing this out. We followed your suggestion and decided to use mesoscale fluctuation throughout the text. To improve clarity, we reformulated this paragraph to start with a clear distinction between microscale turbulence and mesoscale fluctuations.

8. Table 1: Many Gill WindMaster Pro anemometers produced between 2006 and 2015 were affected by a known issue that led to an underestimation of the vertical wind component. See, for example, `https://www.licor.com/support/EddyPro/topics/w-boost-correction.html`. Would it be possible to verify whether this issue affected the instruments used in this study? If so, the bias can be corrected (to some extent) using a straightforward data processing method, as described in the linked resource. Many thanks for pointing this out! Following your comment, we applied the w-boost fix and described it in Sect. 2.1.1.

9. Line 81: Could a brief explanation be provided for the choice of the 6 km area? If this selection is related to internal boundary layers, would it be possible to use a simple analytical model to estimate the internal boundary layer thickness? Garratt (1990) presents several relevant models that might be useful. If such an approach is considered, specifying the roughness length for the sectors of interest would further clarify the reasoning behind the choice. Thank you for the good suggestion. Here, we are interested in the area that influences the turbulence at the measurement height. This can be assessed using the footprint concept. For a rough estimate, we follow Smedman et al. (1999) (see their Appendix A2), who assess the flux footprint for near-neutral conditions and a roughness length typical of the sea ($z_0 = 1.4 \times 10^{-4}$). According to their Eq. A7, the flux contribution within 6 km amounts to over 60% of the total flux footprint influencing measurements at 60 m (see Fig. 1). Following your comment, we added a short explanation to the manuscript.

10. Spike Filtering: It may be beneficial to first apply a flat threshold, such as 65 m/s, which is the upper measurement limit of the Gill sonic anemometer. The reason for this is that spikes can exceed this value, potentially masking other outliers. A possible approach could be: (1) Apply a flat threshold to remove physically unrealistic values. (2) Perform outlier detection using a moving median filter. It is important to use the absolute median deviation (MAD) rather than the absolute mean deviation, as recommended by Leys et al. (2013). It is currently unclear whether the study employs the median or arithmetic mean for outlier detection. The wording "average" suggests the latter, but clarification would be helpful. Thank you for your excellent suggestions. We repeated the filtering following your advice and adopted this approach in the revised version of the manuscript. In the preprint, the arithmetic mean was used to detect outliers, and following Vickers and Mahrt (1997), the procedure of eliminating spikes and recalculating the mean was repeated iteratively until no further spikes were detected. Following your recommendation, we now apply a flat threshold and use the median absolute deviation (MAD) for outlier detection. With this revised implementation, spikes are typically detected in the first iteration, making the despiking algorithm more efficient. The spectra in Figs. 8–10, the extracted Mann parameters, and the sample sizes have changed due to the updated filtering procedure and the changes in response to Specific comments

[Figure]

Figure 1: Cumulative footprint as a function of upwind distance from the measurement location for a measurement height of 60 m, calculated using Eq. (A7) from Smedman et al. (1999). The red lines indicate the 6 km distance used in the manuscript.

8, 11, 12, and 18. Nevertheless, the four key features highlighted in Section 3.3 of the manuscript remain clearly evident in the recalculated spectra. Repeating the analysis and obtaining the same overall conclusions increases our confidence in the robustness of the results.

11. Data Processing and Data Filling: The use of linear interpolation for datasets with 15% missing values (NaNs) raises some concerns, particularly for turbulence studies, where preserving statistical properties is crucial. In atmospheric science, a common threshold for acceptable missing data is around 5%. How would the findings be affected if a stricter threshold were applied, such as 10% or 5% NaNs? Exploring the sensitivity of the results to different thresholds could help assess the robustness of the analysis.

You are right that this point was not formulated clearly in the previous version of the manuscript. We have therefore revised the description for clarity as follows: *"If the percentage of spikes, repeated values, or flickered values is more than 5 % in any of the wind components in a 5 minute segment, the segment is disregarded. To make the best use of the data, disregarded data are filled using data from a time window of 15 minutes before to 15 minute after the center of the disregarded data. Only segments with less than 15% filled data are used to calculate spectra."*

Furthermore, in response to your concern, we reduced the threshold of allowed spikes from 10% to 5%. With this stricter criterion, especially measurement periods in Nuri's back outer eyewall region were excluded, resulting in a substantially reduced sample size for this region. Nevertheless, the enhanced energy at small wavenumbers in Nuri's back inner cyclone region and eyewall region remains clearly evident in the periods with less than 5% spikes. As uncertainty regarding measurement quality cannot be fully excluded, we now explicitly highlight this limitation in the label of Table 3, in the abstract, and in the conclusions. We further state it more clearly in the corresponding descriptions in Sect. 3.3. We refer the reader also to our responses to General comment 6 and Specific comments 22 and 31.

12. Section 2.1.2 – Power Spectra Calculation: The approach described in this section closely resembles Welch's method (Welch, 2003), which is a well-established technique for power spectral estimation. To avoid unnecessary reinvention, it may be beneficial to explicitly state that Welch's method is being used and to reference the appropriate implementation, such as the scipy.signal.welch function in Python or the pwelch function in MATLAB. In this study suggest using around 3 segments with 50% overlapping to reduce uncertainties. Reformulating this section to reflect this could improve clarity and align the methodology with standard signal processing practices. Thank you for this good point, I'll will use the scipy functions in further studies. For this study I reformulated the section sligthly and refer to the Welch's method. Furthermore, I changed to also use 50% overlapping segments in agreement with common practices.

13. Section 2.1.2 – Stationarity Test: Using wind direction change as a criterion for stationarity is a good idea. However, if the goal is to analyze stationary fluctuations specifically, it may also be useful to check stationarity in mean wind speed (first-order stationarity) and variance (second-order stationarity). While this might not be strictly necessary given that the results appear reasonable, performing these additional tests could provide a

more rigorous assessment for future studies. True, with the limitation of the wind direction change stationarity is not guaranteed. However, the aim of our study is not limited to non-stationary periods (see response to General comment 2). We now included a short paragraph to make this clear to the reader.

14. Line 154–155: A clearer formulation might be to state that a key advantage of the Mann model is that the second-order structure of homogeneous atmospheric turbulence in a neutral atmosphere is incorporated using only three parameters. The limited number of parameters is a significant advantage, particularly for wind energy applications, where simplicity and computational efficiency are crucial. Thank you for the suggestion! We followed your recommendation and write now: *"A key advantage of the Mann model is that it incorporates the second-order structure of homogeneous turbulence in a neutral atmosphere using only three parameters."*

15. Line 180–181: The authors raise an important point: the study focuses on stationary turbulence. This distinction should be explicitly mentioned in the abstract, as many readers might initially expect the paper to address non-stationary turbulence for Typhoon winds. We have addressed this in the response to General comment 2.

16. Line 180–181 and lines 224: The claim that significant wind direction changes make the Fourier transform "invalid" is not accurate in my opinion. The Fourier transform remains valid for both stationary and non-stationary signals because it preserves phase information. However, the power spectral density (PSD), which discards phase information, is primarily suited for stationary signals. In strongly non-stationary conditions, such as in the eye and rainband regions of a tropical cyclone, the PSD may not accurately represent the underlying turbulence characteristics. A more precise formulation could be to state that the spectral analysis is not performed in the eye and rainband regions due to the strong non-stationarity of the wind field, which complicates the interpretation of the power spectra. Thanks for this comment. We changed the sentence to: *"The spectral behaviors are not analyzed for the eye and rainband regions because the wind direction changes significantly in these regions, which makes the decomposition of the wind vector into the u and v wind component ambiguous."*

17. Figures 4-7: The figures are well-presented. However, why are the Mann parameters shown only in Figures 4-5 and not in Figures 6-7? If there is a reason for this omission, it would be helpful to clarify it. Thank you for raising this point. We omitted these figures to avoid repetition and to keep the manuscript concise; however, we recognize that this omission may raise questions. Therefore, we now include the Mann parameters corresponding to the other two figures in the Appendix.

18. Figures 4-5: As noted by the authors, some fits have reached the upper boundary for $\Gamma$ and $L$. This suggests that the fitting procedure may not have converged. While such cases can occur, it might be beneficial to disregard those samples (if this has not already been done) or explore whether a better initial guess could improve the fitting. You are right that fits reaching the upper boundary of $\Gamma$ or $L$ may indicate a poor convergence of the fitting procedure. We followed your recommendation and disregarded these cases, when calculating median Mann parameter values. We have clarified this point in the revised manuscript in Sect. 3.2.

19. Mast shadowing: During the passage of a typhoon, wind direction can change dramatically, which may lead to periods where the sonic anemometer data are affected by mast shadowing. To ensure a meaningful analysis, it would be useful to provide the boom orientation and indicate whether and when the data are impacted by mast shadowing. If data are found to be affected, they should be excluded from the analysis. We are aware of this issue. Unfortunately, we don't have the information regarding the boom orientation. We note the possibility that the measurements might be affected by mast shadowing now in connection with Fig. 10, as well as in the discussion and conclusion.

20. Lines 267-268: This is a good observation. As noted in the manuscript, the length scale parameter (L) tends to increase with height above the surface. This trend has been documented in previous studies. Also, it is generally expected that L will increase substantially under convective atmospheric conditions, as discussed in Sathe et al. (2013). Thank you for this helpful comment. We assessed atmospheric stability as described in our response to General comment 1 and in Sect. 4 of this document. Predominantly positive kinematic heat fluxes, indicative of convective conditions, were evident only in Typhoon Nuri's back inner and back eyewall regions. These regions do not coincide with those in which the largest $L$ values were obtained. Nevertheless, we agree that atmospheric stability likely partly influences the observed $L$ values, and we have added a corresponding note to the revised manuscript.

21. Lines 272-273: The variation of $\alpha\epsilon^{2/3}$ is influenced not only by the mean wind speed but also by the variance of the velocity components. The current statement suggests a dependence solely on the mean wind speed, but it would be more accurate to acknowledge both factors. Thank you, we followed your recommendation.

22. Figure 10: Could the secondary peak at $k > 1 \times 10^{-1}$ m$^{-1}$ be related to the first eigenfrequency of the mast? If so, it may be useful to investigate whether structural resonance effects influence the spectral shape. Thank you for this suggestion. Indeed, this provides a plausible explanation for the observed spectral peak. We have incorporated this interpretation into the description of the spectra.

23. Line 355: Good observation. It is possible that many turbulence models used for wind loading underrepresent low-wavenumber variability in the v-component. However, I ignore what could be the physical interpretation of this variability. Boundary layer rolls, as proposed in this study seems to be a reasonable interpretation. We appreciate your thoughts on that!

24. Line 342-344: This statement could be rephrased with more caution. Some of the observed spectral peaks do appear to have a physical origin—for instance, the secondary peak in the $F_v$ -spectrum for the Back Outer region is consistent with trends previously observed in the literature. However, other peaks, such as those for the Front Outer (FO) region, may be influenced by large uncertainties in the lowest-frequency bins due to the use of the modified periodogram method. It is important to remember that power spectral densities (PSDs) are statistical estimates and inherently contain uncertainties that increase as the wavenumber decreases. The spectral peaks observed at $k < 3 \times 10^{-3}$ m$^{-1}$ may be particularly affected by these uncertainties. Ideally, a larger sample size would help determine whether these peaks have a physical basis, but obtaining additional typhoon wind data is understandably challenging. An alternative approach could be to use a greater number of overlapping segments (with at least 50% overlap) in the Welch algorithm to reduce spectral uncertainties and smooth out noise in the lowest frequencies. This is a valid remark, and we acknowledge that the statement may lead to misinterpretation, therefore we removed it from the manuscript.

25. Spectrum shape: Hagupit's spectrum shows a flattened spectral peak, which raises the question of whether ground blockage effects contribute to this flattening. A potential way to assess the impact of ground blockage is to analyze the imaginary part of the cross-spectrum between $u$ and $w$, as suggested in Mann (1994). Thank you for the good point, we include the possiblity to the discussion in Sect. 4.

26. Lines 346-348: The presence of nonzero covariance $v'w'$ in offshore conditions has been documented since the 1980s (see Geernaert (1988)), so this phenomenon is now well established. Maybe these observations should not be described as non-classical turbulence behaviour. Given that the measurements in this study are from coastal or offshore regions, observing significant values in the $F_{vw}$ co-spectrum is expected. It may be useful to mention this as a reminder for readers interested in offshore wind turbine design. Excellent point, we have added it to the manuscript.

27. Lines 350-351: This sentence may need to be rephrased after conducting a more in-depth literature review. While there have been numerous relevant studies, many of them belong to the field of wind engineering, which, somewhat surprisingly, is often overlooked by the wind energy community. This study presents an opportunity to highlight the overlap between these two fields and possibly propose a path toward a unified description of turbulence in tropical cyclones for wind loading on structures. Thank you for this comment. There are indeed several studies assessing tropical cyclone turbulence within the field of bridge and structural engineering, which we now explicitly acknowledge in the Introduction. We have also rephrased the sentence in Lines 350–351 to make it more specific to wind energy applications. In addition, following the reviewer's suggestion, we have added a short paragraph highlighting the overlap between the wind engineering and wind energy communities and discussing the potential for a more unified description of tropical cyclone turbulence.

28. Section 4, paragraph 1: The first paragraph of this section reads more like a conclusion. It could either be removed or merged into Section 5 (Conclusion). Thank you for this suggestion. We agree that the first paragraph of Section 4 reads more like a conclusion. We have therefore moved this paragraph to the Conclusion section, where it better fits the overall structure of the manuscript.

29. Line 359; The Mann model has, to my knowledge, been tested against actual measurements in typhoon winds (see Han et al. (2014)), but only in a superficial manner. In contrast, this study provides a much more detailed analysis, making it both valuable and complementary to previous work. Thank you for sharing this reference! We have weakened the statement accordingly and included the reference into the introduction.

30. Discussion section: It may be useful to discuss how this study fits within the broader framework of stationary vs. non-stationary typhoon wind models. The absence of the Mann model in studies of non-stationary turbulence is reasonable, as it was not designed for such conditions. However, most wind-induced damage to wind turbines

during typhoons is likely caused by extreme, non-stationary gusts. Future studies might explore ways to combine the Mann model with non-stationary models, for example, by incorporating a fluctuating mean wind speed and direction. This could be briefly outlined in the discussion as a potential avenue for further research. Thank you for the suggestion. We have added a paragraph, which addresses the assumption of stationarity at the end of the discussion section.

31. Table 3: This is an informative table, but the comparison of spectral slopes may be misleading, as some slopes in the inertial subrange appear to be spurious. This could mislead readers and practitioners into developing inappropriate wind loading models. Instead of presenting the slope as a result, it may be more appropriate to treat it as a quality check for the data. The last column, which presents the logarithm of spectral peaks, is particularly interesting. It could be beneficial to expand on this aspect. Thank you for this thoughtful comment. We agree that presenting spectral slopes as results may be misleading, and we certainly do not want to encourage inappropriate interpretation or application into turbulence models. In our view, it is not entirely clear, although it is likely, that the positive slopes observed in the inertial subrange are caused by measurement artifacts. Nevertheless, similar features have been interpreted as physical wind characteristics in previous studies (Li et al., 2015). Following your comment, we now include a note into the label of the table, to prevent practitioners from using them directly.

32. The term "excessive energy" may need to be changed into something like "energy larger than predicted by turbulence models designed for a neutral atmosphere". The energy in the u and v components is only excessive if it results from an error. Thank you for the suggestion!, we have changed the formulation to *"larger than predicted by the Mann model."*
Offshore measurements consistently show that these components exhibit higher energy at lower wavenumbers than predicted by the Kaimal model or Mann model. This effect appears to be even more pronounced for typhoon winds, though it may also be influenced by convective conditions. The literature documents a substantial increase in energy of the low-frequency $F_u$ and $F_v$ spectra when the atmosphere becomes slightly convective. This suggests that the observed energy enhancement in the low-wavenumber range is more likely linked to buoyancy effects rather than purely mesoscale fluctuations. This is a valid point. In response to your comment, we examined atmospheric stability during the four typhoon events; a more detailed discussion is provided in our response to General comment 1 and in Sect. 4 of this document. In brief, we find no evidence of strong unstable conditions in the measurements in the outer cyclone regions, i.e., the regions where the Mann model underestimates the spectral energy at low wavenumbers. Thus we are not confident in stating with the current dataset that atmospheric instability is a major cause of the enhanced low-wavenumber spectral energy observed in these regions.

33. Line 374-376: The interpretation that the increased energy at lower wavenumbers is due to mesoscale fluctuations may be criticized. The time and spatial scales used in this study—30-minute averaging with PSD estimation using overlapping segments (potentially reducing it to 10-minute windows)—indicate that the observed structures still fall within the turbulence regime. In fact, in multiple instances within this study, the spectral gap is only just becoming visible, reinforcing the idea that the large eddies observed here are still turbulence—though likely strongly anisotropic or influenced by buoyancy effects. Also, the atmosphere might be slightly convective, which would amplify turbulence through buoyancy-generated eddies, an effect not accounted for in the Mann model. Since the Mann model was developed specifically for neutral conditions, deviations from the model in the low-frequency range under stable or unstable stratification are expected.
In cases where the spectral gap is not visible within a 30 min window (e.g. NURI, Front Outer), the observed spectral behavior is more likely to be convective turbulence than mesoscale fluctuations. Under convective conditions, large turbulence eddies generated by buoyancy can fill the spectral gap, creating a transition zone where turbulence and mesoscale fluctuations may overlap. However, identifying this overlap reliably requires longer averaging times—likely closer to 1 hour or more rather than the 30-minute windows used in this study. If the spectral gap remains invisible even with 1+ hour of data, the observed variability may represent a mixture of large turbulence eddies and mesoscale fluctuations. In such cases, distinguishing between the two might require a turbulence model that explicitly accounts for unstable atmospheric stratification. Thank your for this good point. Following your suggestion, we recalculated the spectra using 90-minute samples in the outer cyclone region, where sufficiently long periods of high-quality measurements are available. These spectra are shown in Fig. 2 of this document. In the figure a weak spectral gap is visible for Hagupit's back outer cyclone, as well as for both the front and back outer cyclone regions of Typhoon Nuri. At the same time, our additional analysis on atmospheric stability suggests that atmospheric stability in the outer cyclone region is not convective (see Sect. 4 of this document). Based on these results, we consider it likely that the larger than predicted spectral

[Figure]

Figure 2: Spectra from the front and back outer cyclone regions of the four typhoon case using 90 minute segments: Median of observed spectra (Solid lines), fitted Mann Model (dashed lines), and estimated maxima of the observed spectra (points).

energy at low wavenumbers in the outer cyclone is at least partly related to mesoscale fluctuations. However, we agree that this interpretation is more uncertain in the inner cyclone and eyewall regions. There, the 30-minute averaging interval is indeed to short to clearly distinguish between turbulent motions and mesoscale variability. In addition, the limited number of sufficiently long samples does not allow to assess the spectral energy with confidence for smaller wavenumbers using 90-minute intervals. We have therefore weakened and clarified the statement in the manuscript to: *"The large energy content at these lower wavenumbers in the outer cyclone region is likely related to significant mesoscale fluctuations observed in previous studies dealing with turbulence under non-tropical cyclone conditions (e.g., De Maré and Mann, 2014; Larsén et al., 2019; Syed and Mann, 2024)."* Finally, to clarify the point regarding overlapping segments: in the manuscript, the spectra are effectively based on 30-minute samples, not 10-minute ones. To increase the overall sample size, 30-minute periods were selected using 15-minute overlapping intervals.

34. Lines 432–440: The statements in this section may be controversial. Aeroelastic codes do not necessarily require spatial series. Since the dynamic response is computed in the time domain by solving the equation of motion, the required input consists of time series, not spatial series. Additionally, this paragraph seems to paraphrase the assumption of stationarity, and I am unsure if it adds substantial value to the paper. A more straightforward approach would be to state that if the assumption of stationarity holds, the methods used in IEC standards can be applied. If it does not hold, a different framework is required. More generally, IEC standards are based on the assumption of stationary, homogeneous, Gaussian, and ergodic turbulence. However, when this assumption is violated—as is often the case for typhoon winds—a different framework must be adopted. This alternative framework is commonly referred to as non-stationary turbulence. Furthermore, non-stationary turbulence frequently exhibits non-Gaussian and non-homogeneous characteristics, as turbulence that deviates from stationarity often displays non-Gaussian properties. This is particularly relevant because non-Gaussian wind loading is known to produce much higher extreme loads than Gaussian wind loading (Gong and Chen, 2014). To my knowledge, this alternative framework has not yet been clearly established in wind energy research. However, it has been extensively studied in wind engineering for bridge design and high-rise building design since the early 2000s. Thank you for your insights and concrete suggestions. The paragraph in the preprint aimed

to acknowledge the difficulty of determining the mean wind speed within a nonstationary tropical cyclone wind field. This complicates the application of the Taylor frozen hypothesis. To clarify this point, we moved the paragraph to Section 2.1.2, where the method is introduced. Furthermore, based on your comment, we added a paragraph addressing the following: 1) the assumption of stationarity, 2) Gaussianity, and 3) opportunities for further work incorporating research on nonstationary tropical cyclone turbulence from the bridge engineering community.

35. Lines 447–451: The critical role of coherence in wind loading has been well recognized since the 1960s (see Davenport (1961)). Therefore, citing additional foundational studies rather than relying solely on Dimitrov et al. as a reference may be a good idea. From the results in Mann (1994) and Cheynet (2019), it appears that Mann's model tends to overestimate the coherence of the u-component at vertical separations. This behavior may also hold for stationary typhoon wind conditions, but further investigation would be needed to confirm this. Previous studies indicate that Mann's model may provide reasonable estimates of turbulence coherence at short lateral separations, but there is a knowledge gap for the case of large lateral separations. Thank you for your helpful summary on this topic. We have incorporated the suggested literature into the paragraph. As this aspect lies beyond the primary focus of the manuscript, the discussion has been kept brief.

36. Conclusions: The conclusion may need to be reformulated and be more closely related to wind energy science. For points 1, it could be a good idea to further elaborate on two key aspects. First, there are large uncertainties at low wavenumbers in the spectra, which should be acknowledged more explicitly. Second, the possible unstable stratification of the atmosphere implies that turbulence eddies generated by buoyancy may be contributing to the observed spectral features. This is not accounted for in the Mann model, as it was developed specifically for neutral atmospheric conditions. The approach using mesoscale simulations could help assess to what extent mesoscale fluctuations and turbulence interact within a 30-minute window, which could be an interesting topic for future research.
First of all, thank you very much for the specific comments and suggestions. These are extremely helpful for further improving the paper! Of course you are right regarding the uncertainties at low wavenumbers in the spectra, therefore we now added: *"We note that estimates of the power spectra based on a limited sample size are inherently subject to uncertainty, particularly at the smallest wavenumbers."* Regarding the stability, following your comments, we have assessed the stability, as described in our answer to General comment 1 and in Sect. 4 of this document. The additional analysis does not suggest obvious unstable conditions in the outer cyclone regions.

Maybe point 2 can be removed from the findings.
We understand your concerns with this point. Therefore, we have included a note on the possible tower fibrations, tower shadowing, or noise and the need for more measurements to improve our understanding.

Instead, the following points could be added:

(a) There are regions within a typhoon where the wind can be described as stationary. In these cases, the Mann model provides a reasonable approximation of the turbulence structure, which is a valuable result.
Thank you for this suggestion. We have incorporated your suggestion by adding the following before the numbered points: *"The analysis revealed that it is difficult to apply the Mann model to the eye and rainbands due to the non-stationarity of the wind field in these regions. In the outer cyclone, inner cyclone, and eyewall regions, the Mann model fit the tropical cyclone power spectra relatively well in some cases. However [...]"*

(b) The spectral peaks of the u- and v-spectra occur at much closer frequencies in this study than predicted by the Mann model. This raises the question of whether this discrepancy is related to a non-neutral atmospheric state or typhoon wind characteristics. This could be explored further.
Thank you also for this suggestion. The point is addressed under item 3 of the list, and a possible relation to boundary-layer rolls is hypothesized below the list. Following your recommendation to further highlight this aspect, we have also added it to the abstract. In addition, we analyzed the atmospheric stability, as described in detail in Sect. 4 of this document. The analysis does not support that there are obvious unstable conditions in the region where this feature is most evident. To keep the conclusions concise, we refrain from expanding the discussion there.

Point 4 might be removed from the key findings. Similar findings have already been reported in non-typhoon conditions. Since this behavior is already well documented, it may not be considered as a "new finding". For

point 5, the behavior described has been well known since the 1980s for offshore wind and is expected to apply to typhoon winds in coastal regions as well. Given its well-documented nature, it may not need to be emphasized as a key new finding.
You are right; these two findings are consistent with previous results under non-tropical cyclone conditions. To emphasize this, we combine the two points and explicitly state that such behavior aligns with findings under non-tropical cyclone conditions. We further take this point out of the abstract.

**3    Technical comments**

1. On the use of $10^{-2.5}\,\mathrm{m}^{-1}$: writing $3 \times 10^{-1}\,\mathrm{m}^{-1}$ is easier to read than $10^{-2.5}\,\mathrm{m}^{-1}$.    Thank you, we changed the notation in the text.

2. Spelling convention: Since Copernicus Publications is a European publisher, it may be preferable to use British English rather than American English for consistency with their style.    Good that you wrote this comment! The English guidelines for WES state the following: *"all standard varieties of English are accepted in order to retain the author's voice. However, the variety should be consistent within each article."*. As the preprint has been written using consistent American English, it is in accordance with the guidelines. Nevertheless, I am thankfull for your comment and will follow your advice for future publications.

3. Page 1, line 9 – Use of "excessive energy": The term "excessive energy" may not be the most appropriate choice, as the model underestimates the observations. A more precise phrasing, such as "more energy than predicted," could better reflect the context.    Thank you for the suggestion, I have followed it.

4. Section 3.3: The title of the subsection may be renamed or adjusted using an question mark "?" Well spotted, thank you.

5. The units of the spectra $kF_i$ , where $i = u, v, w$, should be either in $\mathrm{m\,s^{-1}}$ or $\mathrm{m\,s^{-2}\,Hz^{-1}}$, given that $F$ is expressed in $\mathrm{m^2\,s^{-1}}$ and $k$ is in $\mathrm{m^{-1}}$ . However, Table 3 (and possibly other sections) lists the units as $\mathrm{m^2\,s^{-2}}$, which appears inconsistent. It would be useful to verify and correct the units where necessary.    Thank you for noting the units. There was indeed an error in the units of Fig. 3, which has now been corrected. However, the unit $\mathrm{m^2\,s^{-2}}$ is correct for $kF_i$ and is consistent with other publications. For example, the same units are used in Figs. 7 and 10 in Mann (1994). The unit of the spectra in the wavenumber domain, $F_i$, is $\mathrm{m^3\,s^{-2}}$.

**4    Additional analysis of atmospheric stability**

Following General comment 1, the atmospheric stability of the four typhoons was assessed. For transparency, details of the analysis are provided in the following. To assess atmospheric stability, the stability parameter $\zeta_{loc} = z/L_{loc}$ was calculated, where $z$ denotes the measurement height and $L$ the Obukhov length. Owing to the availability of measurements at $z = 60$ m, the turbulent fluxes were calculated locally at 60 m rather than at the surface, as indicated by the subscript "loc". Concretely, the stability parameter was computed following Eq. 1.

$$\zeta = -\frac{z \; k \; g \; \overline{w'\theta'_v}_{loc}}{u_{*loc}^3 \; T} \tag{1}$$

The local friction velocity $u_{*loc}$ is calculated from sonic anemometer measurements as $u_{*loc} = (\overline{u'w'}_{loc}^2 + \overline{v'w'}_{loc}^2)^{1/4}$. The temperature $T$ is obtained from the sonic anemometers, $k \simeq 0.4$ is the von Kármán constant, and $g$ is the gravitational acceleration. The local kinematic heat flux $\overline{w'\theta'_v}_{loc}$ is estimated from sonic anemometer measurements using the eddy-correlation method and is approximated as $\overline{w'T'}_{loc}$.

Figures 3–6 depict $\zeta_{loc}$, $u_{*loc}$, and $\overline{w'T'}_{loc}$ for the four typhoon cases, evaluated over 10-minute intervals. Across all four typhoons, the scatter in $\zeta_{loc}$ is relatively large, particularly in the outer cyclone regions, where $u_{*loc}$ is comparatively small. However, the magnitude and sign of $\zeta_{loc}$ during typhoon Nuri and Hagupit are consistent with Li et al. (2015) who also analyzed these two typhoons (see their Table 2, noting that their definition of the cyclone regions differs). With the exception of the back eyewall and back inner cyclone regions of typhoon Nuri, the local vertical heat fluxes, $\overline{w'T'}_{loc}$, are predominately small or even weakly negative. This is consistent across all four typhoons in the outer cyclone regions, suggesting that the atmospheric conditions in these cases were likely not convective. Predominantly positive $\overline{w'T'}_{loc}$ fluxes indicating unstable conditions are only found in the back eyewall and back inner region of typhoon Nuri.

**References**

Barthlott, C. and Fiedler, F.: Turbulence structure in the wake region of a meteorological tower, Bound.-Layer Meteorol., 108, 175–190, https://doi.org/10.1023/A:1023012820710, 2003.

Cao, S., Tamura, Y., Kikuchi, N., Saito, M., Nakayama, I., and Matsuzaki, Y.: Wind characteristics of a strong typhoon, J. Wind Eng. Ind. Aerodyn., 97, 11–21, https://doi.org/10.1016/j.jweia.2008.10.002, 2009.

De Maré, M. T. and Mann, J.: Validation of the mann spectral tensor for offshore wind conditions at different atmospheric stabilities, J. Phys.: Conf. Ser., 524, 012106, https://doi.org/10.1088/1742-6596/524/1/012106, 2014.

Gao, Z., Liu, H., Li, D., Yang, B., Walden, V., Li, L., and Bogoev, I.: Uncertainties in temperature statistics and fluxes determined by sonic anemometers due to wind-induced vibrations of mounting arms, Atmos. Meas. Tech., 17, 4109–4120, https://doi.org/10.5194/amt-17-4109-2024, 2024.

Kaimal, J. and Finnigan, J.: Atmospheric boundary layer flows: their structure and measurement, vol. 289, Oxford University Press, 1994.

Larsén, X. G., Larsen, S. E., Petersen, E. L., and Mikkelsen, T. K.: Turbulence characteristics of wind-speed fluctuations in the presence of open cells: A case study, Bound.-Layer Meteorol., 171, 191–212, https://doi.org/10.1007/s10546-019-00425-8, 2019.

Li, L., Kareem, A., Hunt, J., Xiao, Y., Zhou, C., and Song, L.: Turbulence spectra for boundary-layer winds in tropical cyclones: a conceptual framework and field measurements at coastlines, Bound.-Lay. Meteorol., 154, 243–263, https://doi.org/10.1007/s10546-014-9974-7, 2015.

Mann, J.: The spatial structure of neutral atmospheric surface-layer turbulence, J. Fluid Mech., 273, 141–168, https://doi.org/10.1017/S0022112094001886, 1994.

Peña, A.: Østerild: A natural laboratory for atmospheric turbulence, J. Renew. Sustain. Energy, 11, 063302, https://doi.org/10.1063/1.5121486, 2019.

Peña, A., Gryning, S.-E., Mann, J., and Hasager, C. B.: Length scales of the neutral wind profile over homogeneous terrain, J. Appl. Meteorol. Climatol., 49, 792–806, https://doi.org/10.1175/2009JAMC2148.1, 2010.

Smedman, A., Högström, U., Bergström, H., Rutgersson, A., Kahma, K. K., and Pettersson, H.: A case study of air-sea interaction during swell conditions, J. Geophys. Res.: Oceans, 104, 1999JC900213, https://doi.org/10.1029/1999jc900213, 1999.

Syed, A. H. and Mann, J.: A model for low-frequency, anisotropic wind fluctuations and coherences in the marine atmosphere, Bound.-Lay. Meteorol., 190, 1, https://doi.org/10.1007/s10546-023-00850-w, 2024.

Tao, T. and Wang, H.: Modelling of longitudinal evolutionary power spectral density of typhoon winds considering high-frequency subrange, J. Wind Eng. Ind. Aerodyn., 193, 103957, https://doi.org/10.1016/j.jweia.2019.103957, 2019.

Vickers, D. and Mahrt, L.: Quality control and flux sampling problems for tower and aircraft data, J. Atmos. Ocean. Technol., 14, 512–526, https://doi.org/10.1175/1520-0426(1997)014⟨0512:QCAFSP⟩2.0.CO;2, 1997.

Wang, H., Xu, Z., Wu, T., and Mao, J.: Evolutionary power spectral density of recorded typhoons at Sutong Bridge using harmonic wavelets, J. Wind Eng. Ind. Aerodyn., 177, 197–212, https://doi.org/10.1016/j.jweia.2018.04.015, 2018.

Xu, Y. L. and Chen, J.: Characterizing nonstationary wind speed using empirical mode decomposition, J. Struct. Eng., 130, 912–920, https://doi.org/10.1061/(ASCE)0733-9445(2004)130:6(912), 2004.

[Figure]

Figure 3: $\zeta_{loc}$, $u_{*loc}$, and $\overline{w'T'}_{loc}$ for typhoon Hagupit. Typhoon regions further analyzed are marked in panel c): front outer cyclone (FO), front inner cyclone (FI), back inner cyclone (BI), and back outer cyclone (BO).

[Figure]

Figure 4: $\zeta_{loc}$, $u_{*loc}$, and $\overline{w'T'}_{loc}$ for typhoon Nuri. Typhoon regions further analyzed are marked in panel c): front outer cyclone (FO), front inner cyclone (FI), front eyewall (FEW), back eyewall (BEW), back inner cyclone (BI), and back outer cyclone (BO).

[Figure]

Figure 5: $\zeta_{loc}$, $u_{*loc}$, and $\overline{w'T'}_{loc}$ for typhoon Prapiroon. Typhoon regions further analyzed are marked in panel c): front outer cyclone (FO), front inner cyclone (FI), and rainbands (R).

[Figure]

Figure 6: $\zeta_{loc}$, $u_{*loc}$, and $\overline{w'T'}_{loc}$ for typhoon Chanchu. Typhoon regions further analyzed are marked in panel c): front inner cyclone (FI) and back outer cyclone (BO).